# Transporter characterisation reveals aminoethylphosphonate mineralisation as a key step in the marine phosphorus redox cycle

Andrew R. J. Murphy[1], David J. Scanlan [1], Yin Chen [1], Nathan B. P. Adams [2,3], William A. Cadman[2], Andrew Bottrill [1], Gary Bending[1], John P. Hammond [4], Andrew Hitchcock[2], Elizabeth M. H. Wellington [1] & Ian D. E. A. Lidbury [5✉]

The planktonic synthesis of reduced organophosphorus molecules, such as alkylphosphonates and aminophosphonates, represents one half of a vast global oceanic phosphorus redox cycle. Whilst alkylphosphonates tend to accumulate in recalcitrant dissolved organic matter, aminophosphonates do not. Here, we identify three bacterial 2-aminoethylphosphonate (2AEP) transporters, named AepXVW, AepP and AepSTU, whose synthesis is independent of phosphate concentrations (phosphate-insensitive). AepXVW is found in diverse marine heterotrophs and is ubiquitously distributed in mesopelagic and epipelagic waters. Unlike the archetypal phosphonate binding protein, PhnD, AepX has high affinity and high specificity for 2AEP (*Stappia stellulata* AepX $K_d$ 23 ± 4 nM; methylphosphonate $K_d$ 3.4 ± 0.3 mM). In the global ocean, *aepX* is heavily transcribed (~100-fold>*phnD*) independently of phosphate and nitrogen concentrations. Collectively, our data identifies a mechanism responsible for a major oxidation process in the marine phosphorus redox cycle and suggests 2AEP may be an important source of regenerated phosphate and ammonium, which are required for oceanic primary production.

[1] School of Life Sciences, University of Warwick, Gibbet Hill Road, Coventry, UK. [2] Department of Molecular Biology and Biotechnology, University of Sheffield, Sheffield, UK. [3] Nanotemper Technologies GmbH, Flößergasse 4, Munich, Germany. [4] School of Agriculture, Policy, and Development, University of Reading, Earley Gate, Whiteknights, Reading, UK. [5] Department of Animal and Plant Sciences, University of Sheffield, Sheffield, UK. ✉email: l.lidbury@sheffield.ac.uk

Phosphonates are reduced organic phosphorus (P) molecules with a carbon (C)–P bond, as opposed to the more common C-oxygen (O)–P ester bonds found in many other organic P molecules[1]. Phosphonates are synthesised as both primary and secondary metabolites in various bacterial, archaeal and eukaryotic organisms[1–7] where they are incorporated into lipids (phosphonolipids) and glycans (phosphonoglycans)[4,8]. A significant proportion can also be released from the cell to facilitate favourable biotic interactions[9]. Thus, they are ubiquitous in terrestrial and aquatic ecosystems[10–14]. Phosphonates also represent a major fraction of the marine organic P pool[11,15–17] and recent studies have now identified several cosmopolitan marine microorganisms capable of synthesising significant quantities of these compounds[6,7,9,18,19]. Collectively, this synthesis drives a vast global oceanic P redox cycle with reduced P input in the surface ocean estimated to be an order of magnitude greater than (non-anthropogenic) riverine P input[9]. Whilst much attention has focused on the degradation of alkylphosphonates, such as (hydroxy-)methylphosphonate and 2-hydroxyethylphosphonate, as a source of P[16,17,20,21], the potential for aminophosphonates such as 2-aminoethylphosphonate (2AEP) to serve as sources of C and/or nitrogen (N) in the presence of inorganic phosphate (Pi), i.e. in a Pi-insensitive manner, has been neglected. However, emerging evidence suggests that Pi-insensitive 2AEP catabolism occurs in nature[22,23]. Notably, the fact that 2AEP is either absent from, or a minor component of, otherwise phosphonate rich high molecular weight dissolved organic matter (HMW DOM)[16,17], despite its supposed ubiquitous production[9,19,24,25], suggests preferential catabolism of this molecule in comparison to alkylphosphonates.

Unlike the majority of C–O–P monoester bonds, the C–P bond requires specific enzymes to break it, such as the C-P lyase[26,27]. Several 2AEP-specific phosphonate degradation systems have been characterised (Fig. 1a), including the 2AEP transaminase (PhnW) – phosphonoacetaldehyde hydrolase (phosphonatase-PhnX) system[28,29] and the PhnW – phosphonoacetaldehyde dehydrogenase (PhnY)-phosphonoacetate hydrolase (PhnA) system[30–32]. The phosphonate dioxygenase (PhnY*) - phosphohydrolase (PhnZ) system has also been shown to degrade 2AEP[33,34], though at least some homologs of this system are specific to (hydroxy-)methylphosphonate and cannot degrade 2AEP[35,36]. In addition, a gene encoding a recently characterised (R)-1-hydroxy-2-aminoethylphosphonate ammonia lyase (PbfA) is often found in phnWX and phnWAY operons[37], expanding the known repertoire of aminophosphonate degrading capabilites[37]. The C-P lyase, which is a non-specific promiscuous phosphonate degrading enzyme complex, is only induced in response to Pi-starvation, being regulated by the two-component master regulator of the Pi-stress response regulon, PhoBR[24]. In marine surface waters, genes encoding the C-P lyase are enriched in bacterial genomes found in regions typified by low Pi concentrations[20] where they are also heavily expressed[38]. Recent data has shown Pi-insensitive regulation of 2AEP degradation, facilitated by the 2AEP-specific systems, occurs in a few strains of bacteria related to marine Alphaproteobacteria[22] and a terrestrial gammaproteobacterium[23]. In both cases, a major consequence of Pi-insensitive 2AEP degradation was the remineralisation and release of labile Pi[22], due to the greater cellular demand for N over P and the 1:1 N:P stoichiometry of 2AEP.

When analytical methods are not sensitive enough to accurately quantify the concentration and turnover of specific environmental metabolites, screening for the expression of their respective uptake systems becomes an important tool in understanding their in situ cycling[38–41]. To date, only two 2-AEP transport systems have been identified. Both are ATP-binding cassette (ABC) transporters that consist of a periplasmic substrate-binding protein, an ATP-binding domain protein, and a transmembrane permease. The first is located within the C-P lyase operon (phnCDEFGHIJKLMNOP)[42,43] with genes encoding the substrate-binding protein, ATP-binding domain and transmembrane domains designated phnD, phnC, and phnE, respectively. The second is another ABC-transporter PhnSTUV shown by Jiang et al. to complement a C-P lyase knockout mutant of E. coli together with phnWX[28,29]. PhnD has a restricted distribution in seawater with an abundance that is highly correlated with regions of Pi-limitation[20]. However, no 2AEP transporter has been identified in the majority of bacteria possessing PhnWX and PhnWAY systems, which is surprising given the fact that 2AEP is a charged molecule and ubiquitous in marine and terrestrial ecosystems[1,20].

Here, we sought to identify transporters required for 2AEP catabolism in environmental bacteria lacking PhnCDE or PhnSTU. Through combining laboratory-based molecular and genetic analyses with environmental meta-omics, we identified three transporters that have a role in 2AEP uptake and revealed Pi-insensitive 2AEP catabolism is widespread in the global ocean, likely representing a major step in the marine P redox cycle.

## Results

### Pseudomonas putida BIRD-1 possesses a Pi-sensitive 2-aminoethylphosphonate ABC transporter, AepXVW. We recently identified several candidate 2-AEP transporters (ABC-type) in Pseudomonas rhizobacteria that contain the PhnWX phosphonatase but lack both the archetypal PhnCDE transporter and PhnSTUV[44] (Fig. 1a). In Pseudomonas putida BIRD-1 (hereafter BIRD-1), a periplasmic substrate-binding protein associated with one of these putative transporters was induced under Pi-deplete growth conditions in a PhoBR-dependent manner[44]. We hereafter refer to this substrate-binding protein (PPUBIRD1_4925) as 2-aminoethylphosphonate X (AepX). AepX belongs to the same family (pfam13343) as PhnS, iron and sulphate substrate-binding proteins, but is clearly distinct (Coverage = 40%, Identity = 25.09%, $1.1e^{-05}$) (Fig. 1b).

BIRD-1 was capable of growth on 1.5 mM 2AEP as either a sole P, sole N or sole N and sole P source, the latter two resulting in mineralisation of excess Pi that was subsequently exported from the cell (Fig. 1c, Supplementary Fig. 1). Mutagenesis of phnWX confirmed this phosphonatase was essential for 2AEP catabolism under all growth conditions in this bacterium (Supplementary Fig. 2a, b). Next, we investigated if AepX and its corresponding ABC transporter components, the ATP binding domain protein (AepV), and the permease domain protein (AepW) were essential for its growth on 2AEP. Surprisingly, deletion of aepXVW[BIRD] had no effect on growth as a sole N source (Fig. 1c). However, the mutant (ΔaepXVW[BIRD]) had significantly ($p < 0.0001$) attenuated growth on 2AEP as a sole P source (Fig. 1c). The growth defect observed during growth on 2AEP as a sole P source was largely restored by complementing the mutant with a plasmid-encoded native homolog (Fig. 1c). These data suggest that whilst the AepXVW transporter is not essential, it is involved in 2AEP uptake as a sole P source in this bacterium but is not involved in growth as an N source. Therefore, another 2AEP transport system must also exist in this bacterium.

### Identification of two Pi-insensitive 2-aminoethylphosphonate transporters in P. putida BIRD-1. Next, by subjecting the BIRD-1 ΔaepXVW[BIRD] mutant to comparative proteomics, we identified a major facilitator-type transporter, (PPUBIRD1_3129), hereafter referred to as AepP for 2-aminoethylphosphonate permease, whose expression was significantly increased during

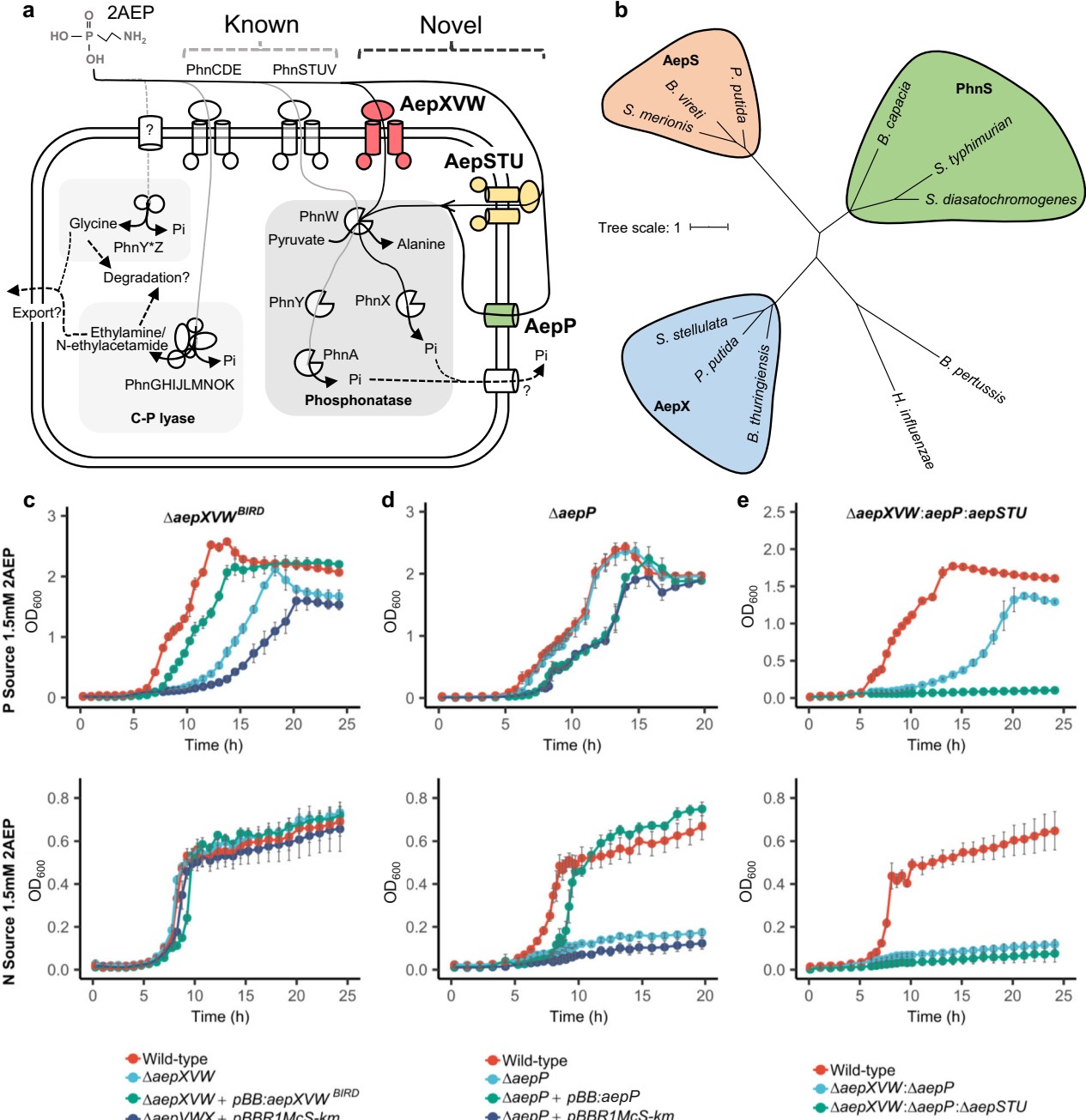

**Fig. 1 2AEP transport and catabolism in *P. putida* BIRD-1. a** Schematic representation of candidate (highlighted in bold and coloured) routes for 2AEP transport, together with existing characterised and putative 2AEP transport routes. Each catabolic system for degradation is highlighted and includes (i) the phosphonatase system comprising a 2AEP-pyruvate transaminase (PhnW) and either a phosphonoacetaldehyde hydrolase (PhnX)[28,29] or a NAD $^+$-dependent phosphonoacetaldehyde dehydrogenase (PhnY)[32] and a phosphonoacetate hydrolase (PhnA)[30,31], and (ii) the PhnY*Z system comprising phosphohydrolase (PhnZ)[33,34] and a 2-oxoglutarate dioxygenase (PhnY*)[33]. In addition, the promiscuous multi-subunit enzyme C-P lyase (PhnGHIJKLMN)[26,27] can also act on 2AEP, as well as akylphosphonates. Pathways found in BIRD-1 are represented by black lines; pathways absent from BIRD-1 are shaded grey. Characterised pathways are shown with solid lines; uncharacterised pathways are shown with dashed lines. Transporters found in BIRD-1 are red if Pi-sensitive, green if Pi-insensitive, and yellow if constitutive. Unknown mechanisms are denoted by a '?'. **b** Phylogenetic tree of AepX, PhnS, and AepS, using the characterised $Fe^{3+}$ substrate-binding protein FbpA from *Haemophilus influenzae* and *Bordetella pertussis* as an outgroup. *P. putida* = *Pseudomonas putida* BIRD-1, *S. stellulata* = *Stappia stellulata*, *B. cepacia* = *Burkholderia cepacia*, *B. vireti* = *Bacillus*, *S. merionis* = *Streptococcus merionis*, *S. typhimurium* = *Salmonella typhimurium*, *S. diasatochromogenes* = *Streptomyces diastatochromogenes*, *B. thuringiensis* = *Bacillus thuringiensis*. **c** Growth (n = 4) of *P. putida* BIRD-1 wild type, Δ*aepXVW*, and the complemented mutant. **d**, Growth (n = 4) of *P. putida* BIRD-1 wild type, Δ*aepP* and the complemented mutant, and (**e**), Growth (n = 4) of *P. putida* BIRD-1 wild type, the double mutant, Δ*aepXVW*:Δ*aepP*, and the 2AEP null transporter mutant, Δ*aepXVW*:Δ*aepP*:Δ*aepSTU*. All strains used 2AEP as a sole P (top panel) or N (bottom panel) source. Error bars denote standard deviation of the mean.

growth on 2AEP as a sole N source (Pi-insensitive) (Supplementary Fig. 3, Supplementary Table 3a). AepP is related to the glycerol-3-phosphate (G3P): Pi antiporter, GlpT[45,46] (identity = 27.75%, 9e$^{-37}$), a member of the organophosphate: Pi antiporter family of major facilitator transporters. AepP shares conserved residues essential for binding Pi and the Pi moiety of G3P[47–50] with GlpT, whereas residues that impact the binding affinity to the glycerol moiety of G3P but not Pi are not conserved[49] (Supplementary Fig. 4). Mutation of *aepP* in either the wild type parental strain (Δ*aepP*, Fig. 1d) or the *aepXVW* mutant (Δ*aepXVW* Δ*aepP*, Fig. 1e) led to an inability to grow on 2AEP as sole N source. Subsequent complementation of Δ*aepP* with its native homolog (Δ*aepP* + pBB:*aepP*) restored growth on 2AEP as a sole N source (Fig. 1d). Interestingly, delayed but significant growth on 2AEP as a P source still occurred in the Δ*aepXVW* Δ*aepP* double mutant, revealing the presence of a third 2AEP transporter in this bacterium.

To identify the unknown 2AEP transporter, we reanalysed our proteomics data (Supplementary Fig. 3). Another substrate binding protein (PPUBIRD1_3891) containing the same pfam domain (pfam13343) as AepX (Fig. 1b), hereafter named AepS, was constitutively synthesised in all growth conditions. In order to uncover the role of AepS in the utilisation of 2AEP as a sole P source, a triple mutant Δ*aepXVW* Δ*aepP* Δ*aepSTU* was generated in BIRD-1 (Fig. 1e). This triple knockout mutant was unable to grow on 2AEP as a sole P source (Fig. 1e), suggesting AepSTU is a functional 2AEP transporter. However, generation of a single Δ*aepSTU* knockout mutant did not affect Pi-sensitive growth compared to the wild type (Supplementary Fig. 2c), suggesting AepXVW is the major transporter involved in Pi-sensitive 2AEP uptake and AepSTU only has an auxiliary role in 2AEP uptake. Despite production of AepS during Pi-insensitive growth (Supplementary Fig. 3), this transporter was unable to facilitate growth on 2AEP as a sole N source in the absence of AepP. Together, these data reveal the presence of three differentially regulated 2AEP transporters in BIRD-1. AepXVW is the primary Pi-sensitive 2AEP transporter with AepSTU having an auxiliary role, whereas AepP is essential for Pi-insensitive growth on 2AEP.

**AepXVW is found in several marine bacteria capable of Pi-insensitive mineralisation and is a functional 2-aminoethylphosphonate transporter.** Using PPUBIRD1_4925 (AepX) as the query, we scrutinised the genomes of several isolates related to marine *Rhodobacteraceae* (*Stappia* spp., *Terasakiella* spp., *Falsirhodobacter* spp.) capable of Pi-insensitive phosphonate catabolism[22]. ORFs encoding orthologs of AepX were identified in the genomes of *Stappia stellulata* DSM 5886 and *Terasakiella pusilla* DSM 6293, in addition to several other marine Roseobacter strains: *Aliiroseovarius crassostreae* (DSM 16950), *Aliiroseovarius sediminilitoris* (DSM 29439), *Shimia marina* (DSM 26895), and *Thalassobius aestuarii* (DSM 15283) (Fig. 2a). We also found an orthologous ORF in the model rhizosphere alphaproteobacterium *Sinorhizobium meliloti* strain 1021 that is capable of 2AEP catabolism via PhnWAY[31] (Fig. 2a). In all cases, ORFs encoding AepXVW were located adjacent to ORFs encoding PhnWAY or PhnWX, strongly suggesting a role in 2AEP transport. *S. stellulata* DSM 5886, *A. crassostreae* DSM 16950, *A. sediminilitoris* DSM 29439, *S. marina* DSM 26895, and *T. aestuarii* DSM 15283 were all capable of growth on 2AEP as either the sole N or P source (Supplementary Table 1, Supplementary Fig. 5). Indeed, both *Aliiroseovarius* strains lack other characterised 2AEP transport and degradation systems (Supplementary Table 2). As previously reported, Pi was exported from cells and accumulated in the medium during growth on 2AEP as

a sole N source (Supplementary Fig. 6a). To confirm that *S. stellulata* AepXVW can take up 2AEP, we complemented the BIRD-1 null 2AEP transporter mutant (Δ*aepXVW*:Δ*aepP*:Δ*aepSTU*) with this transporter fused with the *aepXVW*$^{BIRD}$ promoter (Fig. 2b). This duly restored growth of the triple mutant confirming *S. stellulata* AepXVW is also a functional 2AEP transporter.

In many 2AEP gene clusters, we identified LysR-type regulators, which we refer to as AepR. A homolog of AepR has been shown to be essential for complementation of an *Escherichia coli* Δ*phnHIJKLMNOP* mutant with a Pseudomonad PhnWX[25], implying substrate inducible regulation. Additionally, PhnA activity has been shown to be induced by 2AEP in a marine *Falsirhodobacter* isolate even under nutrient replete conditions[22], though unfortunately no sequenced *Falsirhodobacter* strain possesses an aminophosphonate operon so it is not clear if AepR is responsible for this regulation. BIRD-1 and other Pseudomonads, whose 2AEP operons are often fragmented throughout the genome, possess up to three distinct genes encoding LysR-type regulators (Fig. 2a). In contrast, most *Alphaproteobacteria* only possess a single gene, located upstream of the *aepXVW-phnWAY* operon. These newly identified forms are phylogenetically distinct from either the archetypal PhnR[51] or PalR[52] found in *P. fluorescens* sp. 23F and *Variovorax* sp. PAL2, respectively (Fig. 2c). The three BIRD-1 LysR-like forms were clearly distinct from each other with the AepP-associated form being closely related to the single LysR-type regulator found in *Alphaproteobacteria*.

**AepXVW is highly synthesised in the marine bacterium *Stappia stellulata* during Pi-sensitive and Pi-insensitive growth on 2-aminoethylphosphonate.** In BIRD-1, AepXVW was only involved in Pi-sensitive growth whilst AepP was synthesised during Pi-insensitive and Pi-sensitive metabolism (Fig. 1). *S. stellulata* lacks AepP but is still capable of Pi-insensitive growth and Pi export when grown on 2AEP as a sole N, or sole N and sole P source (Supplementary Fig. 5a). In addition to *aepXVW* and genes encoding the 2AEP degradation system (*phnWAY*), *S. stellulata* also possesses genes (*phnCDEFGHIJKLMNO*) encoding the P-regulated C-P lyase operon and we also confirmed this strain grew on several other alkylphosphonates as sole P source (Supplementary Fig. 5b). Therefore, to determine which transport and degradation systems were upregulated during growth on 2AEP as either a sole N or P source, we performed comparative proteomics. Unlike BIRD-1, AepX was abundantly synthesised during growth on 2AEP as either a sole N or P source, as was PhnWAY and PbfA, the (*R*)-1-hydroxy-2AEP ammonia lyase, whilst the C-P lyase encoded proteins (including PhnD) were not (Fig. 3). Importantly, whilst we detected several general N stress–response proteins induced under Pi-insensitive growth, we did not identify any other potential 2AEP transporters (Supplementary Table 3b). Therefore, unlike in BIRD-1, AepXVW likely represents the major route for 2AEP uptake in this bacterium. These data are consistent with the hypothesis that *S. stellulata* 2AEP catabolism is regulated by a LysR-type regulator solely through substrate-induction, which will be investigated in a future study.

Finally, we determined the substrate specificity of recombinant *S. stellulata* AepX towards 2AEP and other (alkyl)phosphonates using microscale thermophoresis[53,54]. Unlike the relatively promiscuous phosphonate binding protein PhnD[43], AepX was highly specific for 2AEP with a $K_d$ in the nanomolar range (Table 1, Supplementary Fig. 7), consistent with the observation that *aepXVW* is typically co-localised with either *phnWX* or *phnWAY* that encode 2AEP-specific degradation systems (Fig. 2a).

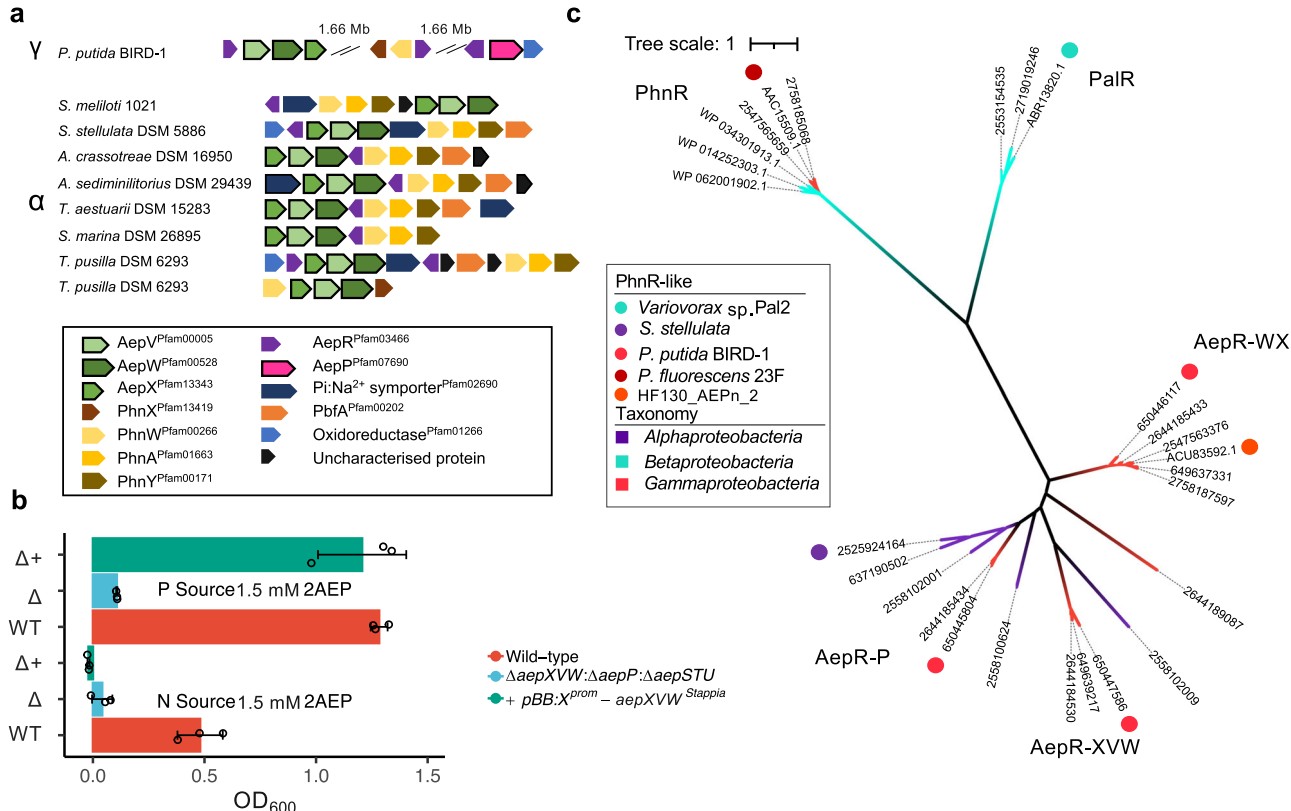

**Fig. 2 Distribution and functional characterisation of AepXVW in marine bacteria. a** Genetic neighbourhoods of *aepXVW* within marine *Alpha-* and terrestrial *Gamma-proteobacteria*. Strains shown are *Pseudomonas putida* BIRD-1, *Sinorhizobium meliloti* 1021, *Stappia stellulata* DSM 5886, *Aliiroseovarius crassostreae* DSM 16950, *Aliiroseovarius sediminilitoris* DSM 29439, *Thalassobius aestuarii* DSM 15283, and *Shimia marina* DSM 26895. ORFs separated on the genome are indicated by breaks with the corresponding gap given in megabases (Mb). **b** Growth of the *P. putida* BIRD-1 triple 2AEP transporter mutant (Δ*aepXVW*:Δ*aepP*:Δ*aepSTU*) complemented with *aepXVW*^*Stappia* concatenated with the promoter region from *aepXVW*^*BIRD* on 2AEP as either a sole N (60 h) or P (48 h) source. Data represent the mean of triplicates cultures. Error bars denote standard deviation. **c** Phylogeny of phosphonate-associated LysR-type regulators. Labels denote IMG/JGI gene IDs or Genbank accession numbers. Tree topology and branch lengths were calculated by maximum likelihood using the LG + G4 model of evolution for amino acid sequences based on 744 sites in IQ-TREE software[84]. Tree Scale represents the number of substitutions per site. A consensus tree was generated using 1000 bootstraps.

***aepX* and *aepP* are found in distantly related and cosmopolitan bacterial taxa.** Using the Integrated Microbial Genomes/Microbiomes from the Joint Genome Institute (IMG/M/JGI) database, we identified ORFs encoding AepX and AepP (but not AepS) in genomes retrieved from both taxonomically divergent isolates as well as single amplified genomes and metagenome-assembled genomes, which revealed an unexpected diversity for these substrate-binding proteins (Fig. 4). For AepX, this included cosmopolitan marine *Alphaproteobacteria* in addition to *Rhodobacteraceae*, marine *Deltaproteobacteria*, and marine *Vibrio* spp. AepX was also found in terrestrial *Betaproteobacteria*, *Firmicutes*, and other gram-positive bacteria (Fig. 4). AepX was partitioned into several subclades, with AepX^*Stappia* and AepX^*BIRD* well separated (Fig. 4). Many taxonomically divergent AepX encoding ORFs were co-localised with ORFs encoding the various 2AEP degradation systems, including the (R)-1-hydroxy-2-aminoethylphosphonate specific PbfA, the C-P lyase, or putative uncharacterised ORFs encoding potentially novel phosphonate catabolic enzymes, supporting a role in 2AEP transport (Fig. 4).

AepP was also found in a wide range of phylogenetically divergent taxa, such as *Acidobacteria* (*Granuliella mallensis*) and *Bacteroidetes* (*Kriegella aquimaris*), *Actinobacteria* (*Streptomyces albulus* CCRC 11814) and *Verrucomicrobia* (*Haloferula* sp. BvORR071 and *Verrucomicrobia sp.* SGGC AC-337 J20) (Supplementary Fig. 6). However unlike AepX, AepP was not found in cosmopolitan marine bacteria. Again, for all of these

strains ORFs encoding AepP were co-localised with ORFs encoding PhnWAY or PhnWX, or putative catabolic enzymes (Supplementary Fig. 8). Notably, AepP was found in fewer marine isolates compared to AepX.

***aepX* gene and transcript abundance is far greater than the archetypal *phnD/phnS* in the global ocean.** Using the TARA oceans OM-RGCv2 + G metagenome (MG) and OM-RGCv2 + T metatranscriptome (MT) datasets[55], we calculated the abundance of *aepX*, *aepS* and *aepP* and compared this with *phnS* and the gene (*phnD*) encoding the archetypal phosphonate transporter, whose gene abundance in seawater was recently calculated[20]. The distribution of markers (*phnJ, phnA, phnX*) for the various phosphonate degradation mechanisms were also analysed (Supplementary Fig. 9). We analysed data from both the epipelagic and mesopelagic zones where phosphonate mineralisation is believed to occur[15]. Across all oceanic sampling sites in both the epipelagic and mesopelagic, *aepX* gene and transcript abundance was significantly greater (MG; post-hoc Dunn's test $z = 10.4$, $p < 0.001$ and $z = 4.8$, $p < 0.001$, respectively) than *phnD* (Fig. 5a, b). On average, in the mesopelagic almost 10% of bacterial cells possess *aepX* whilst only ~0.3% and 0.5% possess *phnD* and *aepP*, respectively (Fig. 5a). The transcript abundance of *aepX* was 40-fold and 140-fold greater than *phnD* in the epipelagic and mesopelagic, respectively (Fig. 5b). The majority of *aepX* sequences were related to the cosmopolitan *Alphaproteobacteria*, although some deltaproteobacterial

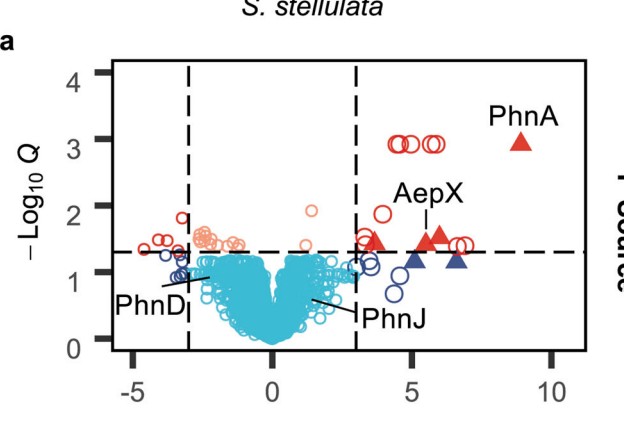

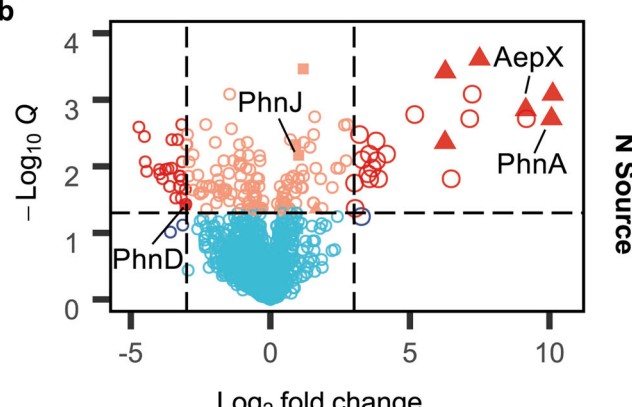

FC~cutoff, 3, q-value~cutoff, 0.05

**Fig. 3 Proteomic analysis of 2AEP-grown *S. stellulata* cells.** Whole-cell protein profiles ($n = 3$) for *S. stellulata* grown using either Pi or 2AEP as sole P source (**a**) or NH$_4$ or 2AEP as the sole N source (**b**). Fold change (FC) represents the difference in Log$_2$ Label Free Quantification (LFQ) values between each treatment and the statistical value on the Y axis is generated from Q values (FDR corrected P values). Members of the *aepXVW-phnWAY* operon are shown as triangles, members of the C-P lyase operon are shown as squares, all other proteins are shown as open circles. Data plotted represents the mean of triplicate cultures. Vertical dashed lines represent an Log$_2$ LFQ difference > −3 or <3. The horizontal dashed line illustrates a cut off for a significant Q value ($p < 0.05$). Sky blue represents proteins showing no significant difference between treatments. Red indicates proteins significantly changing in abundance <−3-fold or >3-fold, respectively. Peach indicates significant changes less than 3-fold in either direction.

**Table 1 Microscale thermophoresis determined dissociation constants ($K_d$) of the *S. stellulata* AepX for selected phosphonate ligands.**

| Ligand | $K_d$ (µM) | |
|---|---|---|
| | AepX Ss | PhnD Ec |
| 2-Aminoethylphosphonate | 0.023 ± 0.004 | >0.050–0.1 |
| Methylphosphonate | 3404.15 ± 280.99 | 1.3–18.4 |
| Ethylphosphonate | 145.96 ± 15.18 | 0.2–1.4 |
| Aminomethylphosphonate | 4490.91 ± 808.24 | 16.6 |

Results are compared against those obtained for *E. coli* PhnD in previous studies[42,65].

sequences fall within this clade (Fig. 4). We confirmed that these abundant environmental sequences were also co-localised with characterised and putative phosphonate degradation genes (Fig. 4). In broad agreement with *aepX*, the cumulative transcription of the two markers *phnA* and *phnX* is significantly greater than the C-P lyase marker *phnJ* (Kruskal–wallis $X^2 = 206.6$, $p < 0.001$), strengthening the observation that 2AEP mineralisation is a major oceanic process (Supplementary Fig. 9). The *phnZ* marker is split into several subclades, with only the original PhnY*Z specific for 2AEP (Supplementary Fig. 10). This 2AEP-specific form was found at very few sites (MG = 9; MT = 2) and in low abundance in both the MG and MT (Supplementary Fig. 10). Homologs related to the two PhnZ clades associated with either methylphosphonate[35,36] or (N)-trimethyl-2-aminoethyl-phosphonate[56] degradation were found at comparable gene and transcript abundances to *phnJ* and significantly lower than *phnA* (Supplementary Fig. 11).

The gene abundances of *aepP*, *aepS* and *phnS* were all significantly lower than both *aepX* and *phnD* (post-hoc Dunn's test $z = 13.1$ and 9.0, $p < 0.001$, $z = 14.5$ and 12.6, $p < 0.001$, and $z = 9.9$ and 8.9, $p < 0.001$ respectively) in the epipelagic, whilst only *aepS* and *phnS* were significantly lower in the mesopelagic (post-hoc Dunn's test $z = 10.1$ and 10.8, $p < 0.001$, and $z = 4.6$ and 5.5, $p < 0.001$ respectively) (Fig. 5a and S12).

Unlike *phnD* and *aepP*, *aepX* abundance was comparable across all oceanic regions within both MG and MT at each depth suggesting 2AEP mineralisation is a ubiquitous process in seawater (Fig. 5c, d). For all sites at each depth, the relative abundance of *aepX* transcripts was always significantly greater (Wilcoxon rank sum $W = 3771$, $p < 0.001$, estimated log$_2$ difference = 2.24 (95% CI 1.97–2.52)) than its own gene abundance. For *phnD*, we observed significantly greater transcript abundance compared to its own gene abundance only in the Mediterranean Sea, a region typified by Pi-depletion (Wilcoxon rank sum $W = 0$, $p < 0.001$, estimated log$_2$ difference = 3.11 (95% CI 1.90–4.42)). Finally, in agreement with previous work[20], *phnD* gene abundance was inversely correlated ($R^2 = 0.340$, $p < 0.001$) with standing stock concentrations of Pi (Fig. 5e), as was *phnD* transcript abundance (Fig. 5f). In contrast, *aepX* and *aepP* gene abundance were positively correlated ($R^2 = 0.098$, $p < 0.001$ and $R^2 = 0.291$, $p < 0.001$, respectively) with Pi concentration (Fig. 5e), whilst no significant relationship between Pi concentration and *aepX*/*aepP* transcript abundance was found (Fig. 5f), suggesting their expression is independent of Pi in seawater globally.

To better understand the parameters controlling 2AEP catabolism in the ocean, we compared both gene and transcript abundance in relation to R*, a measure of N vs P limitation calculated as $[NO_2] + [NO_3] − 16[PO_4]$ (adapted from Smith et al.[57]). As expected, *phnD* gene and transcript abundance were positively correlated with R* ($R^2 = 0.168$, $p < 0.001$ and $R^2 = 0.197$, $p < 0.001$, respectively) (Fig. 5g, h), i.e. regions typified by Pi depletion, though Pi concentration alone was a better predictor of *phnD* gene abundance (Fig. 5e). However, *aepX* and *aepP* gene abundance was (weakly) inversely correlated with R* ($R^2 = 0.029$, $p < 0.05$ and $R^2 = 0.108$, $p < 0.001$, respectively) (Fig. 5g) and no significant relationship was found between R* and *aepX*/*aepP* transcript abundance (Fig. 5h). These data are consistent with the proteomic response of *S. stellulata* under laboratory conditions and suggests AepX is induced in the presence of 2AEP (substrate-inducible) and not in response to nutrient limitation.

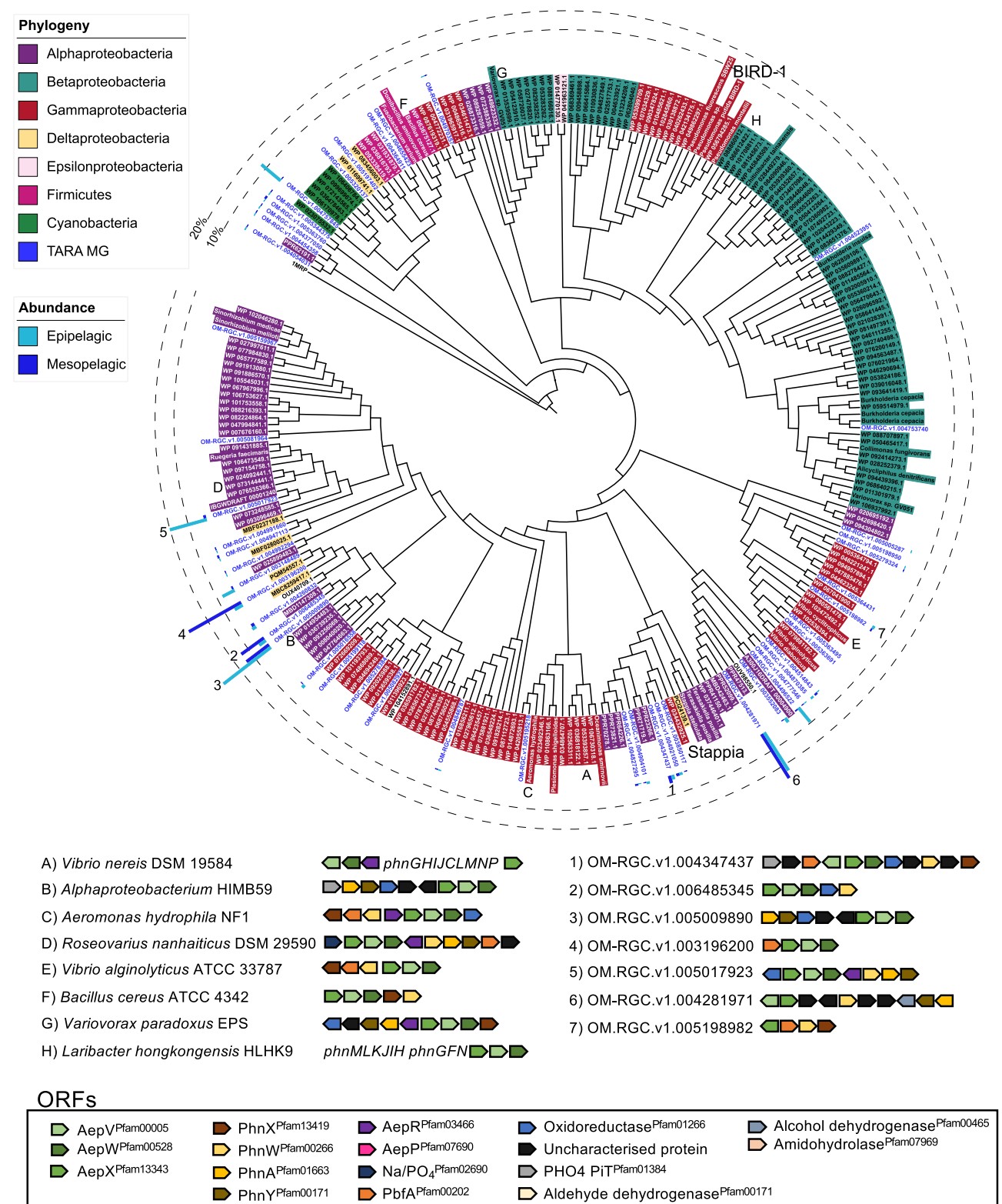

**Fig. 4 Phylogenetic and genomic analyses of AepX in marine and terrestrial bacteria.** Genetic neighbourhoods for selected AepX homologs are presented adjacent to trees. Numbers indicate environmental operational taxonomic units (OTUs) and letters indicate isolates, metagenome-assembled genomes (MAGs) or single-cell amplified genomes (SAGs). Tree topology and branch lengths were calculated by maximum likelihood using the LG + F + I + G4 model of evolution for amino acid sequences based on 744 sites in IQ-TREE software[84]. A consensus tree was generated using 1000 bootstraps. Branches representing isolates or MAGs/SAGs are colour coded based on their phylogenetic affiliation (see legends). Branches and identifiers for representative environmental OTU sequences (clustered at 0.8) retrieved from the TARA Oceans database are highlighted blue. The outer ring denotes the relative abundance of environmental AepX OTUs using the same colour scheme; 10% (dashed line) and 20% (filled line) thresholds are shown for scale. *S. stellulata* DSM 5886 and *P. putida* BIRD-1 AepX are labelled.

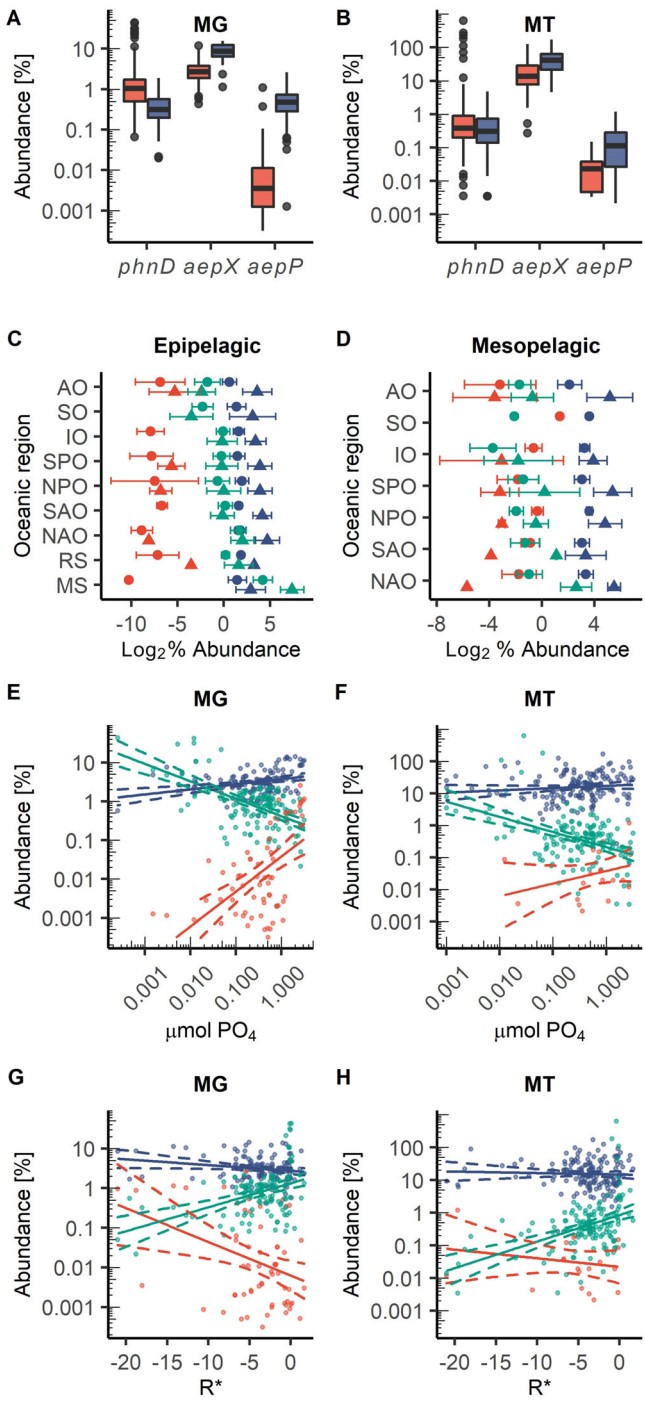

**Fig. 5 Distribution and expression of phosphonate transporter genes in the global ocean.** Abundance (% abundance [gene or transcript] relative to the median abundance [gene or transcript] of 10 single copy core genes) of *phnD*, *aepX*, *aepP* in marine epipelagic (red) and mesopelagic (blue) waters, split by metagenome (MG) (**a**) (epipelagic: *phnD* n = 137, *aepX* n = 137, *aepP* n = 60, mesopelagic: *phnD* n = 43, *aepX* n = 43, *aepP* n = 42, where n equals the number of biologically independent sampling sites where the genes were located), and metatranscriptome (MT) (**b**) (epipelagic: *phnD* n = 148, *aepX* n = 154, *aepP* n = 11, mesopelagic: *phnD* n = 32, *aepX* n = 33, *aepP* n = 18, where n equals the number of biologically independent sampling sites where the transcripts were located). Abundance of *phnD*, *aepX*, *aepP* in MG (circles) and MT (triangles) in epipelagic (**c**) and mesopelagic (**d**) waters, split by oceanic region. *aepP* (red), *phnD* (green), *aepX* (blue). AO Arctic Ocean, SO Southern Ocean, IO Indian Ocean, SPO South Pacific Ocean, NPO North Pacific Ocean, SAO South Atlantic Ocean, NAO North Atlantic Ocean, RS Red Sea, MS Mediterranean Sea. Circles are mean values of Log₂ abundance, error bars represent standard deviation of the mean. Epipelagic (**c**) *phnD* AO MG n = 29, MT n = 26, SO MG n = 3, MT n = 8, IO MG n = 21, MT n = 18, SPO MG n = 25, MT n = 35, NPO MG n = 11, MT n = 20, SAO MG n = 14, MT n = 17, NAO MG n = 16, MT n = 17, RS MG n = 6, MT n = 3, MS MG n = 12, MT n = 7, *aepX* AO MG n = 29, MT n = 28, SO MG n = 3, MT n = 8, IO MG n = 21, MT n = 19, SPO MG n = 25, MT n = 35, NPO MG n = 11, MT n = 20, SAO MG n = 14, MT n = 17, NAO MG n = 16, MT n = 17, RS MG n = 6, MT n = 3, MS MG n = 12, MT n = 7, *aepP* AO MG n = 11, MT n = 3, SO MG n = 0, MT n = 0, IO MG n = 10, MT n = 0, SPO MG n = 13, MT n = 4, NPO MG n = 5, MT n = 2, SAO MG n = 4, MT n = 0, NAO MG n = 11, MT n = 1, RS MG n = 4, MT n = 1, MS MG n = 2, MT n = 0. Mesopelagic (**d**) *phnD* AO MG n = 9, MT n = 7, SO MG n = 1, MT n = 0, IO MG n = 6, MT n = 4, SPO MG n = 9, MT n = 9, NPO MG n = 5, MT n = 5, SAO MG n = 5, MT n = 2, NAO MG n = 8, MT n = 5, *aepX* AO MG n = 9, MT n = 7, SO MG n = 1, MT n = 0, IO MG n = 6, MT n = 4, SPO MG n = 9, MT n = 9, NPO MG n = 5, MT n = 5, SAO MG n = 5, MT n = 2, NAO MG n = 8, MT n = 6, MT n = 7, *aepP* AO MG n = 9, MT n = 6, SO MG n = 1, MT n = 0, IO MG n = 6, MT n = 2, SPO MG n = 9, MT n = 6, NPO MG n = 5, MT n = 2, SAO MG n = 5, MT n = 1, NAO MG n = 7, MT n = 1. The relationship between the standing stock Pi concentration and transporter abundance, analysed by linear regression of $Log_{10}$ Pi concentration and $Log_{10}$ gene/transcript abundance, in the MG (**e**), (*aepX* $R^2 = 0.098$, p = 1.477e-4, *phnD* $R^2 = 0.340$, p = 1.349e-13, *aepP* $R^2 = 0.291$, p = 1.466e-6) and MT (**f**), (*aepX* $R^2 = 0.007$, p = 0.1544, *phnD* $R^2 = 0.203$, p = 4.066e-9, *aepP* $R^2 = 0.058$, p = 0.1581). *aepP* (red), *phnD* (green), *aepX* (blue). The relationship between R*, a measure of N vs P limitation defined as the sum of standing stock nitrate plus nitrite concentration minus 16x standing stock Pi concentration, and transporter abundance, analysed by linear regression of R* and $Log_{10}$ gene/transcript abundance, in the MG (**g**) (*aepX* $R^2 = 0.029$, p = 0.033, *phnD* $R^2 = 0.168$, p = 1.611e-6, *aepP* $R^2 = 0.108$, p = 5.31e-3) and MT (**h**) (*aepX* $R^2 = -0.005$, p = 0.606, *phnD* $R^2 = 0.197$, p = 1.693e-8, *aepP* $R^2 = -0.014$, p = 0.402). *aepP* (red), *phnD* (green), *aepX* (blue). 95% confidence intervals are shown by dashed lines. In (**a**, **b**) data are represented as boxplots, where the middle line is the median and the upper and lower hinges correspond to the first and third quartiles. The upper whisker extends from the upper hinge to the largest value that is no more than 1.5 × IQR (inter-quartile range) from the upper hinge, and the lower whisker extends from the lower hinge to the smallest value that is no further than 1.5 × IQR from the lower hinge. Data beyond the ends of the whiskers are outlying points that are plotted individually. Note: *aepP* transcripts were not detected at numerous sites which is represented by an omission of data points.

## Discussion

Both phosphonate biosynthesis[1,3,5,58] and catabolic[1,23,25,31,59–64] genes are ubiquitous in marine, soil and gut microbiomes, suggesting phosphonate cycling is widespread in nature. In contrast, the uptake of these molecules is comparatively understudied, with only two characterised ABC transport systems confirmed, both of which are linked solely to P-acquisition[28,42,43]. The abundance of these Pi-sensitive transporters in marine systems is not equivalent to the abundance of catabolic genes[20], especially those (*phnWAY*) recently shown to be involved in Pi-insensitive catabolism[22]. This would suggest our knowledge of the microbial uptake of phosphonates, particularly 2AEP, is incomplete. Using transporter expression as a proxy for the cycling of specific nutrients has

significantly advanced our understanding of in situ biogeochemical cycling[39,40,65]. This molecular approach helped resolve the biogenesis of the climate-active gas methane in oxygenated surface waters, driven through the uptake and degradation of methylphosphonate[21,38]. Thus, a gap in mechanistic knowledge on 2AEP metabolism impairs our ability to survey the in situ

cycling of reduced organophosphorus compounds, especially when high-resolution separation of such compounds is difficult[9,66]. Here, identification of Pi-insensitive 2AEP transporters allowed us to develop molecular markers to investigate the cycling of 2AEP on a global scale and compare these with previously characterised Pi-sensitive markers.

To date, whilst several ABC transporters show preference for methylphosphonate and phosphite[53,67], or bind a range of phosphonates with low micromolar or lower $K_d$[43], no ABC transporter showing a strong preference for 2AEP has been identified. Here, we revealed AepX appears to be highly specific for 2AEP and has substantially lower affinity for methylphosphonate or ethylphosphonate than PhnD[43]. The occurrence of aepXVW adjacent to putative phosphonate catabolic genes, and the characterised PbfA[37], does suggest some degree of promiscuous binding, albeit likely to related aminophosphonates. Thus, the molecular mechanisms governing the specificity of AepX towards aminophosphonates warrant further investigation.

Genomic and biochemical analyses have revealed 2AEP and the alkylphosphonates, methylphosphonate and hydroxyethylphosphanate are ubiquitously synthesised in the marine environment in relatively large quantities[1,3,9,25,66]. However, several studies have shown 2AEP is not detected as a significant component of 'semi-labile' DOM whilst alkylphosphonates tend to accumulate[16,17,68]. Collectively, this would suggest 2AEP is more susceptible to microbial mineralisation[22] and thus shorter residence times. Here, we reveal genes for the Pi-insensitive uptake (aepX) and catabolism (phnA) are expressed at significantly higher levels across the global ocean than the Pi-repressible phnD and phnJ, providing a clear mechanism for this phenomenon. Our data also suggest 2AEP is preferentially mineralised independently of both N and P status, explaining why phosphonates are metabolised in regions where Pi or ammonium concentrations are high enough to repress C-P lyase-expression[20,21,69]. Together, this adds further weight to the notion that phosphonates are rapidly cycled between reduced and oxidised forms and are a source of regenerated Pi throughout the water column[9,15,22].

Substrate inducible expression of catabolic genes targeting organic N molecules, irrespective of nutrient status, has previously been shown to drive mineralisation of N and cross feed into surrounding microbial cells[70–73]. Indeed, ammonium mineralisation may also occur if 2AEP, (R)-1-hydroxy-2-aminoethylphosphonate or (N)-trimethyl-2-aminoethylphosphonate are also used as carbon and energy sources, similar to methylamines and quaternary amines[70,71]. In agreement with Chin et al.[22], our proteomic data for S. stellulata and in situ environmental data strongly suggests PhnA and AepX are highly synthesised in a substrate-inducible manner that would facilitate the remineralisation of labile inorganic N and P. Even if ammonium concentration does play a role in the occurrence and regulation of 2AEP degradation genes (i.e. 2AEP is primarily a N source), our combined data clearly demonstrates the potential for the in situ mineralisation of semi-recalcitrant DOP into labile Pi, a mechanism which is important for maintaining biological production in Pi-deplete regions of the ocean[74,75]. It should be noted that the Pseudomonas aepR located adjacent to aepP is most similar to marine aepR; this could explain why AepP was more abundant in the presence of 2AEP (Fig. 1d), although the differences in growth rate and protein abundance suggest substrate induction is not the sole mechanism of regulation in P. putida BIRD-1.

In summary, this study identified three 2AEP transporters in marine and terrestrial bacteria capable of Pi-insensitive 2AEP catabolism. The substrate-binding protein for one of these, AepX, is the most abundant and highly transcribed of the known phosphonate binding proteins in seawater. Therefore, 2AEP mineralisation may represent a major process in the marine organic N and P cycles and a significant source of regenerated labile Pi for oceanic production. This conclusion is strengthened by two key observations: (1) aepX transcription is not repressed by standing stock concentrations of Pi, and (2) extracellular Pi export occurs during Pi-insensitive 2AEP metabolism. Thus, we provide further evidence for the role of low molecular weight phosphonates acting as a P currency between autotrophic and heterotrophic microbes[9].

## Methods

**Bacterial strains and growth conditions.** All strains used in this work were axenic. *Pseudomonas* strains were maintained on Luria Bertani (LB) agar (1.5% w/v) medium at 30 °C. *Stappia stellulata* and the Roseobacter strains were maintained on Marine Broth agar (1.5% w/v) medium at 30 °C. *Pseudomonas* mutants and complemented strains were maintained on similar plates containing the appropriate antibiotic. For all growth and proteomics experiments cultures were grown in an adapted minimal A medium[44] using Na-Succinate (20 mM) as the sole carbon source and, where applicable, 10 mM NH$_4$Cl was added as the sole N source. 2AEP and KH$_2$PO$_4$ were added to a final concentration of 100 μM or 1.5 mM as specified in the text and figure legends. *Pseudomonas* strains were pre-cultured in minimal medium A containing 100 μM Pi and 1.5 mM NH$_4$Cl to ensure adequate growth while minimising the potential for carryover of residual nutrients into experimental cultures. Culture experiments were performed using a FLUOStar Omega 96-well plate reader using Sarstedt 96-well plates incubated at 30 °C, shaking at 200 rpm. For proteomics where 2AEP is used as an N source, *S. stellulata* was grown in modified defined marine ammonium mineral salt (MAMS) media lacking ammonium and with HEPES (4-(2-hydroxyethyl)-1-piperazineethanesulfonic acid) replacing the Pi buffer[71]. For all other growth and proteomics experiments these marine bacteria were grown in Sea Salts media[76], with Pi buffer replaced by HEPES buffer. The method used for quantifying extracellular Pi is detailed in the Supplementary Materials.

**Generation and complementation of *Pseudomonas* mutants.** Mutants were generated and complemented via the protocols outlined in[44,77], detailed descriptions of which are outlined in the Supplementary Materials. A full list of strains, plasmids, and primers used in this study is presented in Supplementary Table 4.

**Proteomics preparation and analysis.** To identify proteins involved in 2AEP uptake and catabolism in *S. stellulata*, total protein ($n = 3$ for each treatment), was retrieved by sampling cell cultures (OD$_{540}$ 0.8–1.0) and pelleting cells (centrifugation at 16200 x g for 5 mins). Cell lysis was achieved via boiling in 100 μl lithium dodecyl sulphate (LDS) buffer (Expedeon) prior to loading 20 μl onto a 4-20% Bis-Tris sodium dodecyl sulphate (SDS) precast gel (Expedeon). For enrichment of the membrane protein fraction of the *P. putida* BIRD-1 ΔaepXVW mutant, we adapted the methods outlined in[44]. A full description of this protocol is outlined in the Supplementary Materials. Gel sections were de-stained with 50 mM ammonium bicarbonate in 50% (v/v) ethanol, dehydrated with 100% ethanol, reduced and alkylated with Tris-2-carboxyethylphosphine (TCEP) and iodoacetamide (IAA), washed with 50 mM ammonium bicarbonate in 50% (v/v) ethanol and dehydrated with 100% ethanol prior to overnight digestion with trypsin. Peptides were extracted and analysed using an Orbitrap Fusion Ultimate 3000 RSLCNano System (Thermo Scientific) in electrospray ionisation mode at the Warwick Proteomics Research Technology Platform.

Resulting tandem mass spectrometry (MS/MS) files were searched against the relevant protein sequence database (*P. putida* BIRD-1, NC_017530.1, *S. stellulata*, GCF_000423715.1) using MaxQuant[78] with default settings and quantification was achieved using Label Free Quantification (LFQ). The proteomics analysis software Perseus (1.6.12)[79] was used to identify differentially expressed proteins based on LFQ values, using a False Discovery Rate (FDR) of 0.01. Identified proteins were retained if they were present in at least two biological replicates within a treatment. Missing (N/A) values were imputed from a normal distribution using the default parameters. Differential expression was identified by two-sample Student's t-test, using an s0 constant of 0.1, or ANOVA, where appropriate.

**Ligand binding affinity of recombinant *S. stellulata* AepX.** Recombinant AepX was over-produced and purified to homogeneity using methods described in[53]. Binding affinity was determined by microscale thermophoresis using a Monolith NT.115 instrument (NanoTemper Technologies GmbT, Germany) following the protocols described in[53,54]. A detailed protocol is outlined in the Supplementary Materials.

**Bioinformatics analyses of *aepXVW/aepP*.** ORFs encoding AepX homologs were identified using the IMG/JGI (https://img.jgi.doe.gov/) 'Customised Homolog Display' search tool. Strains containing homologs were identified (cut-off values:

e[−70], min. identity 40%), preferentially from type strains. In addition, representative strains from soil and marine environments were added to this list (see Supplementary Table 2). Protein sequences were aligned using ClustalOmega[80] and profile Hidden Markov Models (pHMMs) were constructed from these sequences using the hmmbuild function of hmmer 3.3 (http://hmmer.org). The previously characterised *E. coli* K-12 PhnD[81] and the SAR11 clade isolate *Pelagibacter* sp. HTCC7211 PhnD[21] showed surprisingly low sequence homology (BLAST ID 28.46%, query coverage 76%, e-value 2e-25). We therefore, developed two pHMMS for PhnD to reflect this. There was no overlap in environmental sequences retrieved from each search using either pHMM. Therefore, abundance counts for each PhnD form were combined together as a collective PhnD group. These pHMMs were used to search the TARA ocean metagenome (OM-RGC_v2_metaG) and metatranscriptome (OM-RGC_v2_metaT) via the Ocean Gene Atlas web interface[55], using a stringency of 1E[-80]. Sequence abundances were expressed as the average percentage of genomes containing a copy by dividing the percentage of total mapped reads by the median abundance (as a percentage of total mapped reads) of 10 single-copy marker genes[82] for both MG and MT. The pHMMs were used to search the soil MG via a hmmsearch[83] using the same stringency as above. Similarly, abundances were calculated as the average percentage of genomes containing a copy as above.

Phylogenetic analyses were performed using IQ-TREE 2[84] using the following parameters: -m TEST -bb 1000 -alrt 1000. Evolutionary relationships were inferred by maximum-likelihood analysis, and visualised using the Interactive Tree of Life (iTOL) v5.6.3 online platform (https://itol.embl.de/)[85].

**Statistical analysis**. Unless specified above, all statistical analysis was performed using R (version 4.02)[86], within the RStudio programme (version 1.3).

**Reproducibility**. All growth experiments were performed a minimum of two times. Proteomics experiments were performed once.

**Reporting summary**. Further information on research design is available in the Nature Research Reporting Summary linked to this article.

## Data availability

The single-culture proteomic data generated in this study have been deposited in the ProteomeXchange Consortium via the PRoteomics IDEntifications (PRIDE) database under accession code PXD026804 [10.6019/PXD026804]. The TARA Oceans metagenomic and metatranscriptomic data used in this study are available in the European Nucleotide Archive database under accession code PRJEB7988.

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

## Acknowledgements

We thank the Warwick Proteomics Research Facility, namely Dr. Cleidiane Zampronio for her assistance in generating and processing the mass-spectrometry data. This study was funded by the Biotechnology and Biological Sciences Research Council (BBSRC) under project codes BB/L026074/1, BB/T009152/1 and NE/S013539/1 linked to The Soil and Rhizosphere Interactions for Sustainable Agri-ecosystems (SARISA) programme and a Discovery Fellowship (IL) and NERC Environmental 'Omics Synthesis Grant (IL and EW), respectively.

## Author contributions

A.M. and I.L. wrote the manuscript with comprehensive feedback and guidance from Y.C. and D.S. A.M. performed the experimental work, and A.M. and I.L. analysed the data, except for the binding affinity assays, performed and analysed by N.A., W.C., A.H. and I.L. I.L. and A.M. conceptualised and developed the research, respectively. A.B. provided input and support for proteomic data generation and analyses. All authors contributed to revisions of the manuscript.

## Competing interests

The authors declare no competing interests.
