## [Peer Review File · Nature Communications]

REVIEWER COMMENTS

Reviewer #1 (Remarks to the Author):

The distribution, metabolism, and importance of 2AEP in marine ecosystems have long been recognised by the scientific community. Nevertheless, the ecological and biogeochemical role of this natural occurring phosphonate has just begun to be understood. Despite our advance knowledge on the subject, many features on the metabolism of 2AEP remain still undiscovered. For example, the ability of 2AEP-degrading microorganisms to import this phosphonate into the cell, that, however, lack the specific transporters for phosphonates. Therefore, the ability to expand our knowledge on the acquisition and catabolism of phosphonates by microorganisms is a crucial step to comprehend the dynamics of natural occurrence phosphonates in nature fully. Your manuscript describes, in a very structured study and with a well-performed methodology, the discovery of these novel bacterial transporter systems, which will contribute to close the existing gaps in the known metabolism of 2AEP and the role of organophosphonates in the biogeochemical cycle of phosphorus. Therefore, your contribution to this specific topic of interest, which has attracted the attention of many respected scientists since the identification of phosphonates in the seawater column in the early 2000s, will be universally recognised. However, I must say that the current manuscript's title is quite ambitious and does not reflect the extension of the experimental results presented in this work. Therefore, more experimental work is needed at the field level to prove the statement.

Throughout the manuscript, there are a couple of specific gaps in the general knowledge and state of the art about the metabolism of 2AEP that I would like to address in order to increase the quality of your manuscript:

The catabolism of 2AEP by microorganisms is particularly interesting as it can be achieved by different pathways, such as 2-aminoethylphosphonate dioxygenase (PhnY*), phosphonoacetaldehyde hydrolase (phosphonatase – PhnX), and phosphonoacetate hydrolase (PhnA). Interestingly, the operons that contain the genes encoding for either PhnX or PhnA usually contain genes that encode for a 2-AEP transaminase (PhnW) and/or a phosphonoacetaldehyde dehydrogenase (PhnY), which are accessory proteins that participate on the degradation of 2AEP (Agarwal et al. 2014). However, it must be considered that these operons are known to be regulated under either the PhoBR regulatory system related to the Pho regulon or by a LysR-like transcriptional activator responsive to 2AEP, which enable the microorganisms to activate both pathways independently of ambient Pi concentrations (Kulakova et al., 2001, 2009, Cooley et al. 2011, Borisova et al. 2011). Therefore, it is highly recommended that authors should analyse the flanking regions of the operons found in *Pseudomonas putida* BIRD-1, *Stappia stellulata* DMS 5886, and other Pi-insensitive 2AEP-degrading microorganisms to determine if these species possess a LysR type of regulator, as it is known that the majority of phnX operons present in Pseudomonads and phnA in alpha-proteobacteria are associated with this substrate-inducible type or regulator (Villarreal-Chiu et al. 2012).

The phylum proteobacteria is known to excel in accumulating metabolic pathways to degrade natural and xenobiotic compounds, including phosphonates. The existence of different phosphonate degradation pathways or multiple copies of phosphonate degradation genes in members of this genus is well known (*Enterobacter aerogenes*: Lee et al. 1992, *Salmonella typhimurium*: Jiang et al. 1995, *Pseudomonas stutzeri*: White and Metcalf 2004, *Mesorhizobium loti*: Huang et al. 2005). This concept is valid for the presence of multiple copies of the phosphonate transporter gene phnD (*Prochlorococcus marinus*: Feingersch et al. 2012). However, it has been reported that a vast number of bacteria that possessed genes associated with any known phosphonate degradation pathway showed no evidence of possessing the PhnCDE route of phosphonate acquisition (Villarreal-Chiu et al. 2012), which can contribute to exalt the importance of this manuscript. On this regard, it must be mentioned that PhnD has been demonstrated to show the highest affinity to 2AEP. However, this protein shows high specificity to other natural occurring phosphonates as well (Rizk et al. 2006). Therefore, authors must demonstrate those novel proteins AepX and AepP are specific to 2AEP, as stated in the manuscript. Otherwise, authors should confirm that these proteins exhibit a group-

specificity to phosphonates, similarly as PhnD.

Microorganisms have evolved several mechanisms to acquire and conserve P as an adaptation to cope with the ever-changing P concentrations in the environment. These mechanisms are known to be related not just with the Pi levels of the surrounding environment (Pho regulon: Kononova and Nesmeyanova, 2002), but also with nitrogen, iron, oxygen or even light (Dyhrman et al. 2007). A clear example of these mechanisms of acquisition and conservation of P as a response to P and/or N levels, is the ability of microorganisms to accumulate polyhydroxyalkanoates (PHA: Valentino et al. 2015). These are a group of natural polymers accumulated intracellularly as a carbon and energy storage material (Tobin and O'Connor, 2005). PHA production and accumulation has been demonstrated to be widespread among marine microorganisms (Uwamori-Takahashi et al. 2017, Ganapathy et al., 2018). This information is relevant for the present work, as experimentally, PHA production and accumulation by microorganisms can be detected by an increase of turbidity in the culture. This occurs as the PHA internal granules tend to increase against time in response to a nutrient limitation in the presence of excess carbon. This increase of turbidity in the culture can produce false-positive results due to the inability to discriminate between PHA accumulation and biomass production (Acosta-Cortés et al. 2019). On this regard, it is possible that this phenomenon is occurring on the marine bacterial strains tested for 2AEP supplemented as N source, as the utilisation of 2AEP as P is significantly higher to the exceptionally low OD540 obtained for 2AEP cultures supplemented as N source (Table S1). An easy and rapid method to discern the presence of PHA is by visualising it under fluorescent microscopy using a Nile red or Nile blue A 1% staining (Giin-Yu et al. 2014). Demonstrating that *Alliroseovarius* strains do not accumulate PHA when grown on 2AEP supplied as N source, would significantly improve the hypothesis of the existence of a new catabolic pathway for 2AEP.

The bioinformatic analysis of the abundance of phosphonate metabolic pathways among microorganisms have been widely studied (Huang et al. 2005, Villarreal-Chiu et al. 2012). With their new approach, it is recommended that authors should combine their results on the distribution of *aepX* and *aepP* across bacterial taxa and the results of previous studies on the distribution of catabolic pathways. Many potential new transporters or catabolic pathways may be found with this analysis. On the other hand, while authors' approach on the transcript abundance of *aepX* on metagenomic data is valid, it is essential to compare these results with a housekeeping gene in order to normalise the relative abundance of all compared genes against the number of resultants obtained for a housekeeping gene, generally, *recA*, used as a single-copy-per-genome control in distribution and prevalence analyses (Moran et al., 2004).

Finally, references 75, 76, and 77 are missing from the reference list.

References used in this revision:

- Agarwal et al. 2014: *Chem Biol*, 21; DOI: 10.1016/j.chembiol.2013.11.006
Kulakova et al., 2001: *J Bacteriol*, 183: 3268–3275. DOI: 10.1128/JB.183.11.3268-3275.2001
Kulakova et al., 2009: *Microb Biotechnol*, 2: 234–240 DOI: 10.1111/j.1751-7915.2008.00082.x
Cooley et al. 2011: *Microbiology*, 80: 335-340; DOI: 10.1134/S0026261711030076
Borisova et al. 2011: *J Biol Chem*, 286: 22283–22290; DOI: 10.1074/jbc.M111.237735
Villarreal-Chiu et al. 2012: *Front Microbiol*, 3: 19; DOI: 10.3389/fmicb.2012.00019
Lee et al. 1992: *J Bacteriol*, 174: 2501-2510; DOI: 10.1128/jb.174.8.2501-2510.1992
Jiang et al. 1995: *J Bacteriol*, 177: 6411-6421; DOI: 10.1128/jb.177.22.6411-6421.1995
White and Metcalf 2004: *J Bacteriol*, 186: 4730-4739; DOI: 10.1128/jb.186.14.4730-4739.2004
Huang et al. 2005: *J Mol Evol*, 61: 682–690; DOI: 10.1007/s00239-004-0349-4
Feingersch et al. 2012: *ISME J*, 6: 827–834. DOI: 10.1038/ismej.2011.149
Rizk et al. 2006: *Protein Sci*, 15: 1745–1751. DOI: 10.1110/ps.062135206
Kononova and Nesmeyanova, 2002: *Biochemistry*, 67: 184-95; DOI: 10.1023/A:1014409929875
Dyhrman et al. 2007: *Oceanography*, 20: 110-116; DOI: 10.5670/oceanog.2007.54

Valentino et al. 2015: *Water Res*, 77: 49-63; DOI: 10.1016/j.watres.2015.03.016
Tobin and O'Connor, 2005: *FEMS Microbiol Lett*, 253: 111-118; DOI: 10.1016/j.femsle.2005.09.025
Uwamori-Takahashi et al. 2017: *Bioeng*, 4:60; DOI: 10.3390/bioengineering4030060
Ganapathy et al. 2018: *Int J Biol Macromol*, 111: 102-108; DOI: 10.1016/j.ijbiomac.2017.12.155
Acosta-Cortés et al. 2019: *ISME J*, 13: 1497-1505; DOI: 10.1038/s41396-019-0366-3
Giin-Yu et al. 2014: *Polymers*, 6: 706-754; DOI: 10.3390/polym6030706
Moran et al. 2004: *Nature*, 432: 910-913. DOI: 10.1038/nature03170.

Reviewer #2 (Remarks to the Author):

This manuscript describes the discovery and characterisation of several new transporters for 2-aminoethylphosphonate (2AEP), coupled with bioinformatic studies of their prevalence and correlation with environmental factors and phosphonate catabolism genes. The characterisation of the transporters themselves is interesting, but potentially more interesting is the demonstration that these are likely to contribute to phosphate-insensitive metabolism of 2AEP in the marine environment: while this catabolism has been previously shown in isolates, there wasn't clear data on the extent of it in the environment. This is likely to expand the interest in the manuscript beyond microbiologists and into a much broader biogeochemistry/oceanography community, and further supports the recent re-evaluation of the role of reduced phosphorus in the oceans. The experimental approach in the work seems appropriate and is well detailed, and the conclusions are justified. Overall I think this manuscript would make an important contribution to the field.

I have a few core suggestions:

The authors investigate the relationship between the new transporters and PhnA/X/J, and use this to infer that 2AEP-specific mineralisation is a more prevalent process than non-specific phosphonate metabolism via C-P lyase. Is there a reason that PhnZ was omitted from this comparison? It is present in Figure 1 but not examined further.

On line 60 phosphonatases are described as "phosphonate degradation systems", and this word is used throughout the text to refer to the action of PhnA, PhnX, or C-P lyase. The term phosphonatase was coined by La Nauze et al. (*Biochimica et Biophysica Acta Enzymology*, 212[2], 1970) as the trivial name for phosphonoacetaldehyde hydrolase/PhnX specifically, and largely isn't used to mean phosphonate degradation enzymes generally (e.g. see reviews by Horsman and Zechel [2017, *Chemical Reviews*, 117(8)], McGrath et al. [2013, *Nature Reviews Microbiology*, 412], or Peck and van der Donk [2013, *Current Opinion in Chemical Biology*, 17], which all use phosphonatase exclusively to refer to PhnX). It would be more in keeping with the wider phosphonate literature to avoid that term when describing PhnA/C-P lyase.

Line 213 states "aepX transcription was 40-fold and 350-fold greater than phnD in the epipelagic and mesopelagic, respectively (Fig 5B)". The 40-fold number is fine, but Fig. 5B doesn't seem to show a 350-fold difference in aepX and phnD in the mesopelagic, it looks closer to ~130-fold (0.3 vs 40). Is this value correct, or am I misinterpreting something?

The sentence beginning on Line 217 discusses metagenome/transcriptome data for phnA, J and X, but doesn't refer to Figure S8 where this data is shown.

Line 33 states "Collectively, our data identifies a mechanism responsible for the oxidative step in the marine phosphorus redox cycle". Slightly nitpicky, but saying "the oxidative step" implies that this is the only oxidation reaction which phosphorus goes through in the ocean, which isn't correct: aside from phosphonate oxidation systems there are separate enzymes for phosphite and hypophosphite

oxidation present in marine organisms (e.g. Martínez et al., *Environmental Microbiology*, 2011, 14(6), pp1363-77). Perhaps rephrase this along the lines of “for a major oxidation process”?

Line 216: “The majority of *aepX* sequences were related to the cosmopolitan Alphaproteobacteria and Deltaproteobacteria (Fig 4).” Figure 4 appears to show IM-RGC sequences are mostly in the Alpha and Gammaproteobacteria, not the Deltas?

Figure S8F appears to be missing metatranscriptomic data for the Southern Ocean. Figure 5C/D also seem to be missing matching MG/MT data for certain areas (e.g. *aepP* data for the Southern Ocean in 5C, *aepP* MT data for the SAO and MS in 5C, all MT data for SO in 5D). Are these occluded by other symbols or are they missing from the figures/dataset?

Other comments:

Line 55: When arguing that 2AEP is absent from HWM DOP the authors cite article 17 (Sosa et al.). The article in question doesn't state that 2AEP wasn't detected in their sample, only that MPn and 2-HEP were present as the major components along with “minor unidentified phosphonates” (which may have included 2AEP). It may be better to rephrase this sentence to suggest that 2AEP is either absent or present in significantly lower proportions than other phosphonates, thus the ubiquitous synthesis genes would suggest preferential degradation.

In Figure 2A, the colour of the PhnY symbol in the key appears to be darker than the colour used in the diagram itself.

Line 163: “To confirm that *Stappia AepXVW*” should read “To confirm that *Stappia stellulata AepXVW*”

Line 100: “BIRD-1 was capable of growth on 1.5 mM 2AEP as either a sole N, P, or N and P source, the latter resulting in mineralisation of Pi which was subsequently exported from the cell”. Was P export from the cell measured when 2AEP was provided as the sole N source? Previous literature (e.g. citation 22/Chin et al.) would suggest that P export should be observed here as well, and it isn't clear from the text if this didn't occur or just wasn't measured.

Fig. S3: What is the significance of the dashed line at an abundance of 27? This should be stated in the figure legend.

Figure 5's legend is missing a statement identifying the error bars.

Line 316 states “Co-culture experiments were carried out according to the protocol described in60”. It's not clear to me where in the manuscript co-cultures were performed/described, certainly not in the context of the work performed in citation 60. Could the authors clarify this?

Reviewer #3 (Remarks to the Author):

The manuscript “Aminophosphonate mineralisation is a major step in the global oceanic phosphorus redox cycle” uses a combination of involved gene knockout/complementation culture experiments, proteomics, and other multi-OMIC analyses to identify and characterize multiple bacterial aminoethylphosphonate (AEP) transporters. Through their careful analyses they evaluate what controls the expression of these transporters, the fact that the different transporters appear to be involved with either using AEP as a nitrogen (N) or phosphorus (P) source, as well as their ubiquity in both published genomes and global ocean datasets. This is an impressive piece of work and of critical importance to the oceanographic community as it relates to the availability of two essential nutrients

for phytoplankton growth.

That said, I do have several comments on the current manuscript's structure that I feel need to be addressed before publication. The first has to do with the title. The work only focused on AEP, not aminophosphonates in general. While AEP is an example of an aminophosphonate this may be somewhat nit-picking, but I think that Aminoethylphosphonate should replace aminophosphonate in the title. Also, the article focuses much of its discussion on the role in the phosphorus redox cycle, but it's clear from their work that bacteria can use AEP as an N source (and when doing so can release inorganic P), would it be more appropriate to have the title and abstract indicate AEP may be an important N source as well. Global modeling efforts suggest a much larger region of the ocean is N limited than P limited. It is interesting that in meeting N demands, bacteria may release bioavailable inorganic P.

As to the structure of the manuscript. There is an excessive use of abbreviations. I understand that the authors have limited space to tell a large story, but it makes the manuscript very hard to read. Some of this may be alleviated by having the figures (especially figure 1 which has all of the gene abbreviations) closer to the text. The gene name abbreviations are obviously necessary, as are many others, but in thinking about the fact that Nature Communications readership is more broad, it might behoove the authors to question whether it is necessary to introduce the abbreviation for something like MPn and SBP, which are only used a few times in the manuscript and are not common abbreviations outside of a small subset of the field. Also, there's definitely cases where an abbreviation is used that hasn't been introduced and a whole word is used when an abbreviation is in use (and not just at the start of a sentence). I have some other general comments. In the results/discussion of the ubiquity of these transporters in the global datasets, it looks like there's a difference in AepX types (BIRD-1 and Stappia), it looks like it is the Stappia variant that is more abundant in the open ocean. AepX was Pi sensitive in BIRD-1 and BIRD-1 also had AepP, which if I interpret the results correctly was much rarer in both published genomes and the TARA dataset. Did you explore via MAGs or IMG if the organisms that had AepP had a similar variant of AepX as BIRD-1? It seems like an interesting angle to pursue. When discussing the MG and MT data, all focus is on the role of Pi in potentially controlling abundance/expression. AEP is a potential N source. Considering there are some weak but significant inverse correlations with R^* for gene abundance (though not expression), it seems like this should at least be explored in the discussion. Finally, I would suggest some alterations to the bold statement in the summary paragraph (lines 293-294). I didn't see MG/MT data for AepV and AepW (though I might have missed this), so I would limit this to AepX. Also, given that this work has expanded the known number of phosphonate transporters by 3 and there still could be others out there, I would qualify that AepX is the most abundant of the known phosphonate transporters.

Before I list the final more copy-edit style comments, I want to reiterate that I think the author make a compelling case that this is an important new discovery that will require a re-assessment of the phosphorus (and nitrogen) cycle in the surface ocean.

Other specific comments:

Line 47-49: The way this is phrased, it almost suggests that this is only important in response to increased anthropogenic P loading.

Line 53: Pi has not been introduced yet.

Line 58-60: The opening sentence of this paragraph is really awkwardly written and hard to follow.

Line 67: after strains, add "of bacteria" (assuming that's what you mean)

Line 78: whose? Sentence is confusing as written. For clarity, replace "whose" with "with an" and insert a "that" after abundance

Line 80: PhnWAY not introduced, add "that" after fact

Line 82-83: This statement about transporters being superb molecular tools for investigating in-situ cycling of metabolites, while being a true statement, seems out of place/unconnected.

Lines 94-97: This sentence is very confusing as written with too many clauses so that it's hard to sort out which clause is related to which statement.

Line 101: Other statements reference Pi being exported whenever AEP is the N source (i.e., AEP as just N source or as N and P source). What that not the case in BIRD-1 (or did the authors not check for Pi release when BIRD-1 was growing on AEP as an N source with added Pi? This should be made clearer (as should the later statements).

Line 179: N has been introduced as abbreviation for nitrogen

Line 180: What or where is table S3? Is it the "dataset" that is attached? If so, please make that clear and add a table legend so that the reader knows what they're looking at.

Line 188: Statement as written implies that this is absent from the Rhodobacteriaceae. Do you mean in addition to?

Line 316: What do you mean by co-culture experiments? Please provide a bit more detail.

Line 325: Table S4 is missing strain information. Also, as a general comment, I assume that all of the strains used in this work were axenic. The authors should explicitly state this somewhere.

Methods: General comment. There are a lot of abbreviations in the methods of things that are somewhat standard/well known to a biologist, but probably should still be spelled out given the audience for Nature Communications (HEPES, LDS, PCR).

Figure 3 Legend: This legend is hard to follow and doesn't explain all of the components of the figure. What do the colors mean? What do the dashed lines mean?

REVIEWER COMMENTS

We thank all three reviewers for their expert remarks. We have now performed several new experiments and analyses in accordance with their comments which we hope meets with their approval and allows acceptance of this manuscript in *Nature Communications*, as follows:

- 1) Ligand-binding affinity assays of recombinant *S. stellulata* AepX
- 2) Enumeration of cells for *Rosobacter* strains growing on 2AEP as either an N or P source.
- 3) Pi efflux quantification in *P. putida* BIRD-1 cultures where 2AEP is a sole N source
- 4) Bioinformatics analysis of the distribution of PhnZ, the key marker of the 2AEP oxidative degradation pathway.
- 5) Bioinformatics analysis of the LysR-type regulators located adjacent to various 2AEP utilisation gene clusters.

Please note that in addition to the point-by-point responses, we have slightly amended the methods section. This now includes methods for quantifying extracellular phosphate as well as the production, purification and characterisation of recombinant AepX cloned from *Stappia stellulata* and associated ligand binding assays. We have also moved some of the details on protocols to the Supplementary Materials in order to save space.

We have also amended the abstract to include the new binding affinity data and other suggestions, whilst adhering to the 150 max word limit.

Reviewer #1 (Remarks to the Author):

The distribution, metabolism, and importance of 2AEP in marine ecosystems have long been recognised by the scientific community. Nevertheless, the ecological and biogeochemical role of this natural occurring phosphonate has just begun to be understood. Despite our advance knowledge on the subject, many features on the metabolism of 2AEP remain still undiscovered. For example, the ability of 2AEP-degrading microorganisms to import this phosphonate into the cell, that, however, lack the specific transporters for phosphonates. Therefore, the ability to expand our knowledge on the acquisition and catabolism of phosphonates by microorganisms is a crucial step to comprehend the dynamics of natural occurrence phosphonates in nature fully. Your manuscript describes, in a very structured study and with a well-performed methodology, the discovery of these novel bacterial transporter systems, which will contribute to close the existing gaps in the known metabolism of 2AEP and the role of organophosphonates in the biogeochemical cycle of phosphorus. Therefore, your contribution to this specific topic of interest, which has attracted the attention of many respected scientists since the identification of phosphonates in the seawater column in the early 2000s, will be universally recognised. However, I must say that the current manuscript's title is quite ambitious and does not reflect the extension of the experimental results presented in this work. Therefore, more experimental work is needed at the field level to prove the statement.

We acknowledge that our manuscript lacks environmental field data, and have thus amended the title accordingly to highlight this fact:

'Transporter characterisation reveals aminoethylphosphonate mineralisation as a key step in the marine phosphorus redox cycle'

We have largely retained the latter part of the original title since we do believe that our new data on the binding affinity of AepX combined with our originally presented omics data presents a very compelling argument for the high turnover of 2AEP in seawater and hence a key step in the marine phosphorus redox cycle. Clearly, there is now an urgent need to use refined analytical techniques to obtain field-based process data.

Throughout the manuscript, there are a couple of specific gaps in the general knowledge and state of the art about the metabolism of 2AEP that I would like to address in order to increase the quality of your manuscript:

The catabolism of 2AEP by microorganisms is particularly interesting as it can be achieved by different pathways, such as 2-aminoethylphosphonate dioxygenase (PhnY*), phosphonoacetaldehyde hydrolase (phosphonatase – PhnX), and phosphonoacetate hydrolase (PhnA). Interestingly, the operons that contain the genes encoding for either PhnX or PhnA usually contain genes that encode for a 2-AEP transaminase (PhnW) and/or a phosphonoacetaldehyde dehydrogenase (PhnY), which are accessory proteins that participate on the degradation of 2AEP (Agarwal et al. 2014). However, it must be considered that these operons are known to be regulated under either the PhoBR regulatory system related to the Pho regulon or by a LysR-like transcriptional activator responsive to 2AEP, which enable the microorganisms to activate both pathways independently of ambient Pi concentrations (Kulakova et al., 2001, 2009, Cooley et al. 2011, Borisova et al. 2011). Therefore, it is highly recommended that authors should analyse the flanking regions of the operons found in *Pseudomonas putida* BIRD-1, *Stappia stellulata* DMS 5886, and other Pi-insensitive 2AEP-degrading microorganisms to determine if these species possess a LysR type of regulator, as it is known that the majority of phnX operons present in Pseudomonads and phnA in alpha-proteobacteria are associated with this substrate-inducible type or regulator (Villarreal-Chiu et al. 2012).

We thank the reviewer for this insightful comment. The LysR-family of regulators are actually highlighted in Figure 2A. However, we acknowledge that the link between our labelled 'AepR' and LysR-like was not clearly presented. We do have a large amount of data on the regulation of the 2AEP transporters and PhnWX in *P. putida*. We initially had some of this data presented in this manuscript. However, the regulation is very complex and involves the three master regulators of the C, N and P stress responses, CbrAB, NtrBC and PhoBR, respectively. Therefore, we removed this for simplicity since we are performing a more comprehensive analysis of the regulation, which is the focus of a follow-on paper.

However, we have now added a phylogenetic comparison of the LysR-like regulators found in 2AEP operons (Figure 2C). This demonstrates that *P. putida* possess three distinct LysR-like regulators for AepXWV (AepR-XWV), AepP (AepR-P) and PhnWX (Aep-WX) that are divergent from either PhnR or PalR. *S. stellulata* and *S. meliloti* possess just the one upstream of *aepXWV-phnWAY*, which is most similar to the AepP-associated LysR-type. Thus, it is likely that these regulators have evolved to perform divergent functions. To our knowledge, given the divergence between PhnR and aminophosphonate operon LysR-type regulators, the only direct experimental evidence regarding their function comes from Martinez *et al.* 2010, where a *lysR* gene was shown to be essential for complementing an *E. coli* Δ *phnHIJKLMN*OP mutant with a *Pseudomonas* PhnWX system. We note, though, that substrate induction via these LysR-like regulators in no way precludes regulation through other mechanisms – indeed the data we refer to above demonstrates this.

We have added a paragraph starting line 163:

'In many 2AEP gene clusters, we identified LysR-type regulators, which we refer to as AepR. A homolog of AepR has been shown to be essential for complementation of an *Escherichia coli* Δ *phnHIJKLMN*OP mutant with a Pseudomonad PhnWX²⁵, implying substrate inducible regulation. Additionally, PhnA activity has been shown to be induced by 2AEP in a marine *Falsirhodobacter* isolate even under nutrient replete conditions²², though unfortunately no sequenced *Falsirhodobacter* strain possesses an aminophosphonate operon so it is not clear if AepR is responsible for this regulation. BIRD-1 and other Pseudomonads, whose 2AEP

operons are often fragmented throughout the genome, possess up to three distinct genes encoding LysR-type regulators (Fig 2A). In contrast, most *Alphaproteobacteria* only possess a single gene, located upstream of the *aepXVW-phnWAY* operon. These newly identified forms are phylogenetically distinct from either the archetypal PhnR or PalR found in *P. fluorescens* sp. 23F and *Variovorax* sp. PAL2, respectively (Fig 2C). The three BIRD-1 LysR-like forms were clearly distinct from each other with the AepP-associated form being closely related to the single LysR-type regulator found in *Alphaproteobacteria*.

As well as a sentence starting at line 192:

'These data are consistent with, the hypothesis that *S. stellulata* 2AEP catabolism is regulated by a LysR-type regulator solely through substrate-induction, which will be investigated in a future study.'

The phylum proteobacteria is known to excel in accumulating metabolic pathways to degrade natural and xenobiotic compounds, including phosphonates. The existence of different phosphonate degradation pathways or multiple copies of phosphonate degradation genes in members of this genus is well known (*Enterobacter aerogenes*: Lee et al. 1992, *Salmonella typhimurium*: Jiang et al. 1995, *Pseudomonas stutzeri*: White and Metcalf 2004, *Mesorhizobium loti*: Huang et al. 2005). This concept is valid for the presence of multiple copies of the phosphonate transporter gene *phnD* (*Prochlorococcus marinus*: Feingersch et al. 2012). However, it has been reported that a vast number of bacteria that possessed genes associated with any known phosphonate degradation pathway showed no evidence of possessing the PhnCDE route of phosphonate acquisition (Villarreal-Chiu et al. 2012), which can contribute to exalt the importance of this manuscript. On this regard, it must be mentioned that PhnD has been demonstrated to show the highest affinity to 2AEP. However, this protein shows high specificity to other natural occurring phosphonates as well (Rizk et al. 2006). Therefore, authors must demonstrate those novel proteins AepX and AepP are specific to 2AEP, as stated in the manuscript. Otherwise, authors should confirm that these proteins exhibit a group-specificity to phosphonates, similarly as PhnD.

Given this referee's very interesting comment regarding the binding specificity of our newly characterised transporters we have now performed ligand-binding interaction assays for the *Stappia stellulata* AepX using microscale thermophoresis (MST). Our data (Table 1, Figure S6) highlights a key finding revealing that AepX has very high affinity for 2AEP (K_d 23±4 nM) compared to a much lower affinity for ethylphosphonate (145±15 µM), methylphosphonate (K_d 3.4±0.28 mM) and aminomethylphosphonate (4.4±0.8 mM). This finding is consistent with the association of *aepXVW* with 2AEP-specific degradation genes. Thus, AepX differs from the *E. coli* PhnD in so much that it has much greater specificity towards a single phosphonate (2AEP).

We have not provided any data for AepP for two reasons: 1) AepP is a membrane protein that is not trivial to over-express, purify and subsequently assay, issues which are only magnified by the current pandemic. 2) Importantly, this transporter is far less abundant in the ocean and thus its characterisation would not significantly affect the narrative of this paper, unlike AepX.

We have now added the AepX binding data to the results in the *S. stellulata* proteomics section starting line 195 and in the abstract:

'Finally, we determined the substrate specificity of recombinant *S. stellulata* AepX towards 2AEP and other (alkyl)phosphonates using microscale thermophoresis^{53,54}. Unlike the relatively promiscuous phosphonate binding protein PhnD⁴³, AepX was highly specific for 2AEP with a K_d in the nanomolar range (Table 1, Supplementary Fig. 7), consistent with the observation that *aepXVW* is typically co-localised with either *phnWX* or *phnWAY* that encode 2AEP-specific degradation systems (Fig. 2A).'

&

'Unlike the archetypal phosphonate binding protein, PhnD, AepX has high specificity for 2AEP (*Stappia stellulata* AepX K_d 23±4 nM; methylphosphonate K_d 3.4±0.3 mM).'

We have also added the following sentences in the discussion starting line 296:

'To date, whilst several ABC transporters show preference for methylphosphonate and phosphite^{53,67}, or bind a range of phosphonates with low micromolar or less K_d ⁴³, no ABC transporter showing a strong preference for 2AEP has been identified. Here, we revealed AepX appears to be highly specific for 2AEP and has substantially lower affinity for methylphosphonate or ethylphosphonate than PhnD⁴³. The occurrence of *aepXVW* adjacent to putative phosphonate catabolic genes, and the characterised PbfA³⁷, does suggest some degree of promiscuous binding, albeit likely to related aminophosphonates. Thus, the molecular mechanisms governing the specificity of AepX towards aminophosphonates warrant further investigation.'

Microorganisms have evolved several mechanisms to acquire and conserve P as an adaptation to cope with the ever-changing P concentrations in the environment. These mechanisms are known to be related not just with the Pi levels of the surrounding environment (Pho regulon: Kononova and Nesmeyanova, 2002), but also with nitrogen, iron, oxygen or even light (Dyhrman et al. 2007). A clear example of these mechanisms of acquisition and conservation of P as a response to P and/or N levels, is the ability of microorganisms to accumulate polyhydroxyalkanoates (PHA: Valentino et al. 2015). These are a group of natural polymers accumulated intracellularly as a carbon and energy storage material (Tobin and O'Connor, 2005). PHA production and accumulation has been demonstrated to be widespread among marine microorganisms (Uwamori-Takahashi et al. 2017, Ganapathy et al., 2018). This information is relevant for the present work, as experimentally, PHA production and accumulation by microorganisms can be detected by an increase of turbidity in the culture. This occurs as the PHA internal granules tend to increase against time in response to a nutrient limitation in the presence of excess carbon. This increase of turbidity in the culture can produce false-positive results due to the inability to discriminate between PHA accumulation and biomass production (Acosta-Cortés et al. 2019). On this regard, it is possible that this phenomenon is occurring on the marine bacterial strains tested for 2AEP supplemented as N source, as the utilisation of 2AEP as P is significantly higher to the exceptionally low OD₅₄₀ obtained for 2AEP cultures supplemented as N source (Table S1). An easy and rapid method to discern the presence of PHA is by visualising it under fluorescent microscopy using a Nile red or Nile blue A 1% staining (Giin-Yu et al. 2014). Demonstrating that *Alliroseovarius* strains do not accumulate PHA when grown on 2AEP supplied as N source, would significantly improve the hypothesis of the existence of a new catabolic pathway for 2AEP.

We thank the reviewer for their thoughts here and have now determined biomass increase by enumerating growth via colony forming units (c.f.u). This clearly demonstrates that cells are actively catabolising and assimilating the nitrogen to facilitate proliferation as opposed to simply accumulating excess carbon (Figure S5).

The lower biomass observed when 1.5 mM 2AEP is used as the sole N source compared to when 1.5 mM 2AEP is used as the sole P source is due to the higher cellular demand for N relative to P. Given the 1:1 ratio of N:P in 2AEP, one would expect a lower final biomass. We have now added controls using 1.5 mM ammonium as the sole N source and you can clearly see comparable OD₅₄₀ and c.f.u counts between the controls and 2AEP-grown cells.

The bioinformatic analysis of the abundance of phosphonate metabolic pathways among microorganisms have been widely studied (Huang et al. 2005, Villarreal-Chiu et al. 2012). With

their new approach, it is recommended that authors should combine their results on the distribution of *aepX* and *aepP* across bacterial taxa and the results of previous studies on the distribution of catabolic pathways. Many potential new transporters or catabolic pathways may be found with this analysis.

We apologise that this was not clearer in our manuscript. Our analysis also included markers of the catabolic enzymes (*phnA*, *phnJ* and *phnX*). We have now added a line to reference Fig S8:

'The distribution of markers (*phnJ*, *phnA*, *phnX*) for the various phosphonate degradation systems were also analysed (Fig S8)'

We found that *PhnA* was the most abundant catabolic gene also. However, we did not identify any other putative transporters, other than *AepXVW*, when screening the neighbourhoods of some of these gene clusters. We also scrutinised the neighbourhoods of several *AepX* and *AepP* genes and did indeed identify several putative catabolic genes for phosphonate degradation, shown in Figure 4.

We have now amended the results section to acknowledge there were novel catabolic genes located in these *AepX/AepP* neighbourhoods:

'Many taxonomically divergent *AepX* ORFs were co-localised with ORFs encoding the various 2AEP degradation systems, the C-P lyase, or putative uncharacterised ORFs encoding potential novel phosphonate catabolic enzymes, supporting a role in 2AEP transport (Fig 4).'

Also:

'Again, for all of these strains ORFs encoding *AepP* were co-localised with ORFs encoding *PhnWAY* or *PhnWX*, or putative catabolic enzymes (Fig S6).'

Also:

'We confirmed that these abundant environmental sequences were also co-localised with characterised and putative phosphonate degradation genes (Fig 4).'

On the other hand, while authors' approach on the transcript abundance of *aepX* on metagenomic data is valid, it is essential to compare these results with a housekeeping gene in order to normalise the relative abundance of all compared genes against the number of resultants obtained for a housekeeping gene, generally, *recA*, used as a single-copy-per-genome control in distribution and prevalence analyses (Moran et al., 2004).

We fully agree. We think the reviewer has missed our statement concerning normalisation methods which can be found in either the figure legend or methods section. We have already normalised all marker gene data (MG or MT) to the median gene abundance (for MG) or transcript count (for MT) of ten housekeeping genes. Hence, there is a value greater than 100% in some MT samples. Indeed, this further demonstrates very high expression of *AepX*.

(Figure 5):

'Abundance (Log2 % abundance [gene or transcript] relative to the median abundance [gene or transcript] of 10 single copy core genes) of *phnD*, *aepX*, *aepP* in marine epipelagic (red) and mesopelagic (blue) waters, split by metagenome (MG) (A), and metatranscriptome (MT).'

Methods:

'Sequence abundances were expressed as the average percentage of genomes containing a copy by dividing the percentage of total mapped reads by the median abundance (as a percentage of total mapped reads) of 10 single-copy marker genes⁸² for both MG and MT'

Finally, references 75, 76, and 77 are missing from the reference list.
We apologise for this. These are now added.

References used in this revision:

We have now added the appropriate references suggested by yourself. Others we already had in the manuscript.

Agarwal et al. 2014: Chem Biol, 21; DOI: 10.1016/j.chembiol.2013.11.006 Included
Kulakova et al., 2001: J Bacteriol, 183: 3268–3275. DOI: 10.1128/JB.183.11.3268-3275.2001 Included
Kulakova et al., 2009: Microb Biotechnol, 2: 234–240 DOI: 10.1111/j.1751-7915.2008.00082.x Included
Cooley et al. 2011: Microbiology, 80: 335-340; DOI: 10.1134/S0026261711030076 Included
Borisova et al. 2011: J Biol Chem, 286: 22283–22290; DOI: 10.1074/jbc.M111.237735 Included
Villarreal-Chiu et al. 2012: Front Microbiol, 3: 19; DOI: 10.3389/fmicb.2012.00019 Included
Lee et al. 1992: J Bacteriol, 174: 2501-2510; DOI: 10.1128/jb.174.8.2501-2510.1992
Jiang et al. 1995: J Bacteriol, 177: 6411-6421; DOI: 10.1128/jb.177.22.6411-6421.1995 Included
White and Metcalf 2004: J Bacteriol, 186: 4730-4739; DOI: 10.1128/jb.186.14.4730-4739.2004 Included
Huang et al. 2005: J Mol Evol, 61: 682–690; DOI: 10.1007/s00239-004-0349-4
Feingersch et al. 2012: ISME J, 6: 827–834. DOI: 10.1038/ismej.2011.149. Included
Rizk et al. 2006: Protein Sci, 15: 1745–1751. DOI: 10.1110/ps.062135206. Included
Kononova and Nesmeyanova, 2002: Biochemistry, 67: 184-95; DOI: 10.1023/A:1014409929875 Included
Dyhrman et al. 2007: Oceanography, 20: 110-116; DOI: 10.5670/oceanog.2007.54 Included
Valentino et al. 2015: Water Res, 77: 49-63; DOI: 10.1016/j.watres.2015.03.016
Tobin and O'Connor, 2005: FEMS Microbiol Lett, 253: 111-118; DOI: 10.1016/j.femsle.2005.09.025
Uwamori-Takahashi et al. 2017: Bioeng, 4:60; DOI: 10.3390/bioengineering4030060
Ganapathy et al. 2018: Int J Biol Macromol, 111: 102-108; DOI: 10.1016/j.ijbiomac.2017.12.155
Acosta-Cortés et al. 2019: ISME J, 13: 1497-1505; DOI: 10.1038/s41396-019-0366-3
Giin-Yu et al. 2014: Polymers, 6: 706-754; DOI: 10.3390/polym6030706
Moran et al. 2004: Nature, 432: 910-913. DOI: 10.1038/nature03170.

Reviewer #2 (Remarks to the Author):

This manuscript describes the discovery and characterisation of several new transporters for 2-aminoethylphosphonate (2AEP), coupled with bioinformatic studies of their prevalence and correlation with environmental factors and phosphonate catabolism genes. The characterisation of the transporters themselves is interesting, but potentially more interesting is the demonstration that these are likely to contribute to phosphate-insensitive metabolism of 2AEP in the marine environment: while this catabolism has been previously shown in isolates, there wasn't clear data on the extent of it in the environment. This is likely to expand the interest in the manuscript beyond microbiologists and into a much broader biogeochemistry/oceanography community, and further supports the recent re-evaluation of

the role of reduced phosphorus in the oceans. The experimental approach in the work seems appropriate and is well detailed, and the conclusions are justified.

Overall I think this manuscript would make an important contribution to the field.

I have a few core suggestions:

The authors investigate the relationship between the new transporters and PhnA/X/J, and use this to infer that 2AEP-specific mineralisation is a more prevalent process than non-specific phosphonate metabolism via C-P lyase. Is there a reason that PhnZ was omitted from this comparison? It is present in Figure 1 but not examined further.

We initially omitted PhnZ since there are a variety of forms. To address this comment we have now constructed a phylogeny of all PhnZ-like sequences (Fig S9). Characterised forms can be involved in either methylphosphonate or 2AEP catabolism and there is substantial sequence variation even within variants of known function. As such, our hmm detects all forms of PhnZ, including the N-trimethyl-2-aminoethylphosphonate specific TmpB. In addition to these three forms, several PhnZ-like ORFs are frequently found associated with PhnWAY/XW and even C-P lyase operons, inflating the abundance of this gene, but with no real evidence as to their function. We also scrutinised the TARA dataset for PhnY*, the first enzyme in this degradation pathway, but found it to be in very low abundance (Fig S10). This further indicates diversification in the function of PhnZ. All of this is really exciting, but beyond the scope of our current paper which specifically focuses on 2AEP metabolism.

We have added this sentence to the introduction starting line 54:

'The phosphonate dioxygenase (PhnY*) phosphohydrolase (PhnZ) system has also been shown to degrade 2AEP^{33,34}, though at least some homologs of this system are specific to (hydroxy-)methylphosphonate and cannot degrade 2AEP^{35,36}.'

We have also added this small paragraph to the results section starting line 243:

'The *phnZ* marker is split into several subclades, with only the original PhnY*Z specific for 2AEP (Fig S9). This 2AEP-specific form was found at very few sites (MG = 9; MT=2) and in low abundance in both the MG and MT (Fig S9). Homologs related to the two PhnZ clades associated with either methylphosphonate or N-Trimethyl-2-aminoethylphosphonate degradation were found at comparable gene and transcript abundances to *phnJ* and significantly lower than *phnA* (Fig S9).'

On line 60 phosphonatasases are described as "phosphonate degradation systems", and this word is used throughout the text to refer to the action of PhnA, PhnX, or C-P lyase. The term phosphonatasase was coined by La Nauze et al. (Biochimica et Biophysica Acta Enzymology, 212[2], 1970) as the trivial name for phosphonoacetaldehyde hydrolase/PhnX specifically, and largely isn't used to mean phosphonate degradation enzymes generally (e.g. see reviews by Horsman and Zechel [2017, Chemical Reviews, 117(8)], McGrath et al. [2013, Nature Reviews Microbiology, 412], or Peck and van der Donk [2013, Current Opinion in Chemical Biology, 17], which all use phosphonatasase exclusively to refer to PhnX). It would be more in keeping with the wider phosphonate literature to avoid that term when describing PhnA/C-P lyase.

Thank you for this comment and we agree. We initially thought it would simplify the terminology for a general audience, but fully accept it is better to stay in keeping with the current literature. We have corrected throughout the manuscript.

Line 213 states "aepX transcription was 40-fold and 350-fold greater than phnD in the epipelagic and mesopelagic, respectively (Fig 5B)". The 40-fold number is fine, but Fig. 5B doesn't seem to show a 350-fold difference in aepX and phnD in the mesopelagic, it looks closer to ~130-fold (0.3 vs 40). Is this value correct, or am I misinterpreting something?

Thank you kindly for spotting this. We have checked the raw data and it is ~140-fold (0.3 v 42).

Sentence changed to:

'40-fold and 140-fold greater than.....'

The sentence beginning on Line 217 discusses metagenome/transcriptome data for *phnA*, *J* and *X*, but doesn't refer to Figure S8 where this data is shown.

Again, thank you very much for spotting this. We have now added the appropriate citation:

'...major oceanic process (Fig S8).'

Line 33 states "Collectively, our data identifies a mechanism responsible for the oxidative step in the marine phosphorus redox cycle". Slightly nitpicky, but saying "the oxidative step" implies that this is the only oxidation reaction which phosphorus goes through in the ocean, which isn't correct: aside from phosphonate oxidation systems there are separate enzymes for phosphite and hypophosphite oxidation present in marine organisms (e.g. Martínez et al., *Environmental Microbiology*, 2011, 14(6), pp1363-77). Perhaps rephrase this along the lines of "for a major oxidation process"?

Thank you and agreed. Changed to:

'responsible for a major oxidation process in the...'

Line 216: "The majority of *aepX* sequences were related to the cosmopolitan Alphaproteobacteria and Deltaproteobacteria (Fig 4)." Figure 4 appears to show IM-RGC sequences are mostly in the Alpha and Gammaproteobacteria, not the Deltas?

This statement refers to sequences pulled out from the TARA oceans dataset, which are coloured blue in the figure. Some of the abundant OTUs, denoted by the outer ring bars are related to the SAR324 cluster.

Figure S8F appears to be missing metatranscriptomic data for the Southern Ocean. Figure 5C/D also seem to be missing matching MG/MT data for certain areas (e.g. *aepP* data for the Southern Ocean in 5C, *aepP* MT data for the SAO and MS in 5C, all MT data for SO in 5D). Are these occluded by other symbols or are they missing from the figures/dataset?

There is no mesopelagic MT site, and only a single mesopelagic MG site, in the Southern Ocean in the TARA dataset. If a gene is absent from an oceanic region it is not shown. If a gene is only detected at a single location within an oceanic region no error bars are shown. This only occurs for *aepP*, the least abundant transporter. We have amended the figure legends to make this clear:

'Note: *AepP* transcripts were not detected at numerous sites; represented by an omission of data points.'

Other comments:

Line 55: When arguing that 2AEP is absent from HWM DOP the authors cite article 17 (Sosa et al.). The article in question doesn't state that 2AEP wasn't detected in their sample, only that MPn and 2-HEP were present as the major components along with "minor unidentified phosphonates" (which may have included 2AEP). It may be better to rephrase this sentence to suggest that 2AEP is either absent or present in significantly lower proportions than other phosphonates, thus the ubiquitous synthesis genes would suggest preferential degradation.

We agree. Changed to:

'Notably, the fact that 2AEP is either absent from, or a minor component of, otherwise phosphonate rich high molecular weight dissolved organic matter (HMW DOM)^{16,17}.'

Additionally, in the discussion:

'However, several studies have shown 2AEP is not detected as a significant component of 'semi-labile' DOM whilst alkylphosphonates tend to accumulate^{16,17,65}.'

In Figure 2A, the colour of the PhnY symbol in the key appears to be darker than the colour used in the diagram itself.

Thank you for spotting this. This has now been changed.

Line 163: "To confirm that *Stappia AepXVW*" should read "To confirm that *Stappia stellulata AepXVW*"

Thank you. We have changed to:

'To confirm that *S. stellulata AepXVW*...'

Line 100: "BIRD-1 was capable of growth on 1.5 mM 2AEP as either a sole N, P, or N and P source, the latter resulting in mineralisation of Pi which was subsequently exported from the cell". Was P export from the cell measured when 2AEP was provided as the sole N source? Previous literature (e.g. citation 22/Chin et al.) would suggest that P export should be observed here as well, and it isn't clear from the text if this didn't occur or just wasn't measured.

We have now repeated this experiment with *P. putida* to match the same data as we collected for *S. stellulata*. We have amended Figure S1 accordingly. The data now clearly shows mineralisation occurs even when exogenous phosphate is present.

We have amended the sentence in the results to make this statement clearer:

'BIRD-1 was capable of growth on 1.5 mM 2AEP as either a sole P, N or N and P source, the latter two resulting in mineralisation of Pi which was subsequently exported from the cell (Fig 1C, Fig S1).'

Fig. S3: What is the significance of the dashed line at an abundance of 27? This should be stated in the figure legend.

We initially added this here as a visual aid. However, we have now removed this as there is nothing statistically relevant about this line.

Figure 5's legend is missing a statement identifying the error bars.

Thank you kindly for spotting this. We have now added:
'Error bars denote standard deviation of the mean.'

Line 316 states "Co-culture experiments were carried out according to the protocol described in60". It's not clear to me where in the manuscript co-cultures were performed/described, certainly not in the context of the work performed in citation 60. Could the authors clarify this?

Many apologies for this statement which is a relic of an earlier version of the manuscript. This sentence has now been removed from the revised manuscript.

Reviewer #3 (Remarks to the Author):

The manuscript "Aminophosphonate mineralisation is a major step in the global oceanic phosphorus redox cycle" uses a combination of involved gene knockout/complementation culture experiments, proteomics, and other multi-OMIC analyses to identify and characterize multiple bacterial aminoethylphosphonate (AEP) transporters. Through their careful analyses they evaluate what controls the expression of these transporters, the fact that the different transporters appear to be involved with either using AEP as a nitrogen (N) or phosphorus (P) source, as well as their ubiquity in both published genomes and global ocean datasets. This is an impressive piece of work and of critical importance to the oceanographic community as it relates to the availability of two essential nutrients for phytoplankton growth.

That said, I do have several comments on the current manuscript's structure that I feel need to be addressed before publication. The first has to do with the title. The work only focused on AEP, not aminophosphonates in general. While AEP is an example of an aminophosphonate this may be somewhat nit-picking, but I think that Aminoethylphosphonate should replace aminophosphonate in the title. Also, the article focuses much of its discussion on the role in the phosphorus redox cycle, but it's clear from their work that bacteria can use AEP as an N source (and when doing so can release inorganic P), would it be more appropriate to have the title and abstract indicate AEP may be an important N source as well. Global modelling efforts suggest a much larger region of the ocean is N limited than P limited. It is interesting that in meeting N demands, bacteria may release bioavailable inorganic P.

Thank you for this comment. Indeed, in response to reviewer 1 we have obtained new data showing AepX binds to 2AEP with high specificity compared to ethylphosphonate, methylphosphonate or aminomethylphosphonate. As mentioned in our response to reviewer 1 we believe that our new data on the binding affinity of AepX combined with our originally presented omics data presents a very compelling argument for the high turnover of 2AEP in seawater and hence a key step in the marine phosphorus redox cycle. Hence, we feel justified to retain that aspect of the original title (but indeed changing aminophosphonate for aminoethylphosphonate).

Whilst we thank the referee for highlighting that 2AEP can be used as an N source we do not think it appropriate to add this aspect to the title given we have no data to compare its use with other organic N sources.

Our justification for focussing on the P redox cycle is because we directly compare all phosphonate utilisation genes which likely represent the major pathways for oxidation of reduced phosphorus. Given the very high expression of AepX and PhnA in seawater compared to PhnD, PhnJ and PhnZ, our data provides a clear mechanism for one of the major steps in this oxidative pathway. In contrast, we do not compare AepX expression to other N cycling genes. This is beyond the scope of the current manuscript and would require a very detailed analysis of all pathways linked to the degradation of e.g. methylated amines, quaternary amines, polyamines, amino acids, etc. Thus, at the moment we believe it is too speculative to state 2AEP mineralisation is an important part of the N cycle, or carbon for that matter. However, we do acknowledge that the role in N cycling should be explored in the discussion and our data does provide a ubiquitous molecular mechanism for the regeneration of ammonium. We have thus significantly amended the discussion section on 2AEP mineralisation as follows:

'Substrate inducible expression of catabolic genes targeting organic N molecules, irrespective of nutrient status, has previously been shown to drive mineralisation of N and cross feed into surrounding microbial cells⁶⁸⁻⁷¹. Indeed, ammonium mineralisation may also occur if 2AEP,

(*R*)-1-hydroxy-2-aminoethylphosphonate or (*N*)-trimethyl-2-aminoethylphosphonate are also used as carbon and energy sources, similar to methylamines and quaternary amines^{68,69}. In agreement with Chin *et al.*²², our proteomic data for *S. stellulata* and *in situ* environmental data strongly suggests PhnA and AepX are highly synthesised in a substrate-inducible manner, that would facilitate the remineralisation of labile inorganic N and P. Even if ammonium concentration does play a role in the occurrence and regulation of 2AEP degradation genes (i.e. 2AEP is primarily a N source), our combined data, clearly demonstrates the potential for the *in situ* mineralisation of semi-recalcitrant DOP into labile Pi, a mechanism which is important for maintaining biological production in Pi-deplete regions of the ocean^{72,73}. It should be noted that the *Pseudomonas* AepR located adjacent to AepP is most similar to marine AepR. This could explain why AepP was more abundant in the presence of 2AEP (Fig 1D), although the differences in growth rate and protein abundance suggest substrate induction is not the sole mechanism of regulation in *P. putida* BIRD-1.'

In addition, we have also recognised the N cycle in the abstract and have also amended the abstract at line 26:

'2AEP may be an important source of regenerated phosphate and ammonium, which are required for oceanic primary production.'

As to the structure of the manuscript. There is an excessive use of abbreviations. I understand that the authors have limited space to tell a large story, but it makes the manuscript very hard to read. Some of this may be alleviated by having the figures (especially figure 1 which has all of the gene abbreviations) closer to the text. The gene name abbreviations are obviously necessary, as are many others, but in thinking about the fact that Nature Communications readership is more broad, it might behoove the authors to question whether it is necessary to introduce the abbreviation for something like MPn and SBP, which are only used a few times in the manuscript and are not common abbreviations outside of a small subset of the field.

Thank you and we do agree. We have changed many of these now to keep in full.

Also, there's definitely cases where an abbreviation is used that hasn't been introduced and a whole word is used when an abbreviation is in use (and not just at the start of a sentence).

Thank you for spotting. We have now checked the manuscript and amended where appropriate.

I have some other general comments. In the results/discussion of the ubiquity of these transporters in the global datasets, it looks like there's a difference in AepX types (BIRD-1 and *Stappia*), it looks like it is the *Stappia* variant that is more abundant in the open ocean. AepX was Pi sensitive in BIRD-1 and BIRD-1 also had AepP, which if I interpret the results correctly was much rarer in both published genomes and the TARA dataset. Did you explore via MAGs or IMG if the organisms that had AepP had a similar variant of AepX as BIRD-1? It seems like an interesting angle to pursue.

Thank you for this thoughtful comment. We found no correlation between the AepX variant and the possession of AepP. Only *Pseudomonads* have both AepX and AepP, but other AepP homologs are found in strains lacking AepX.

We have now added binding data and analysis of the LysR-type regulators as requested by reviewer 1. It has been noted that the LysR-type activator found in alphaproteobacterial AEP operons is more similar to the LysR-type regulator associated with AepP. This may point towards a divergent role for *Pseudomonas*-like compared to other AepX homologs. On the other hand, Betaproteobacteria in the same clade as the *Pseudomonas* possess a LysR-like regulator that is more similar to the AepXVW.

As stated to reviewer 1, we have two other papers detailing both structure-function relationships and regulation of these transporters, the latter of which is very complicated.

When discussing the MG and MT data, all focus is on the role of Pi in potentially controlling abundance/expression. AEP is a potential N source. Considering there are some weak but significant inverse correlations with R^* for gene abundance (though not expression), it seems like this should at least be explored in the discussion.

We agree. Similar to your previous comment above, we have now amended the section discussing substrate-inducible expression to add more weight to the notion that 2AEP may result in remineralised ammonium and/or serve as a nitrogen source.

‘Substrate inducible expression of catabolic genes targeting organic N molecules, irrespective of nutrient status, has previously been shown to drive mineralisation of N and cross feed into surrounding microbial cells⁶⁸⁻⁷¹. Indeed, ammonium mineralisation may also occur if 2AEP, (R)-1-hydroxy-2-aminoethylphosphonate or (N)-trimethyl-2-aminoethylphosphonate are also used as carbon and energy sources, similar to methylamines and quaternary amines^{68,69}. In agreement with Chin *et al.*²², our proteomic data for *S. stellulata* and *in situ* environmental data strongly suggests PhnA and AepX are highly expressed in a substrate-inducible manner, that would facilitate the remineralisation of labile inorganic N and P. Even if ammonium concentration does play a role in the occurrence and regulation of 2AEP degradation genes (i.e. 2AEP is primarily a N source), our combined data, clearly demonstrates the potential for the *in situ* mineralisation of semi-recalcitrant DOP into labile Pi, a mechanism which is important for maintaining biological production in Pi-deplete regions of the ocean^{72,73}. It should be noted that the *Pseudomonas* AepR located adjacent to AepP is most similar to marine AepR. This could explain why AepP was more abundant in the presence of 2AEP (Fig 1D), although the differences in growth rate and protein abundance suggest substrate induction is not the sole mechanism of regulation in *P. putida* BIRD-1.’

Finally, I would suggest some alterations to the bold statement in the summary paragraph (lines 293-294).

We have amended the concluding summary statement as follows:

‘One of these, AepX, is the most abundant and highly transcribed of the known phosphonate transporters in seawater. Therefore, 2AEP mineralisation may represent a major process in marine organic N and P cycles and a significant source of regenerated labile Pi for oceanic production.’

I didn't see MG/MT data for AepV and AepW (though I might have missed this), so I would limit this to AepX. Also, given that this work has expanded the known number of phosphonate transporters by 3 and there still could be others out there, I would qualify that AepX is the most abundant of the known phosphonate transporters.

Two valid comments. We have now changed accordingly.

Before I list the final more copy-edit style comments, I want to reiterate that I think the author make a compelling case that this is an important new discovery that will require a re-assessment of the phosphorus (and nitrogen) cycle in the surface ocean.

Thank you very much.

Other specific comments:

Line 47-49: The way this is phrased, it almost suggests that this is only important in response to increased anthropogenic P loading.

This has been changed to the following to emphasise that the comparison is with riverine input:

'Collectively, this synthesis drives a vast global oceanic phosphorus redox cycle with reduced phosphorus input in the surface ocean estimated to be an order of magnitude greater than (non-anthropogenic) riverine phosphorus input⁹.'

Line 53: Pi has not been introduced yet.

Thank you for spotting this. We have now changed this to make the sentence clearer:

'...sources of C and/or nitrogen (N) in the presence of inorganic phosphate (Pi), i.e. in a Pi-insensitive manner, has been neglected.'

Line 58-60: The opening sentence of this paragraph is really awkwardly written and hard to follow.

Thank you. We have now changed to:

'Unlike the majority of C-O-P monoester bonds, the C-P bond requires specific enzymes to break it, such as the C-P lyase^{26,27}.'

Line 67: after strains, add "of bacteria" (assuming that's what you mean)

Thank you for this. Changed to:

'occurs in a few strains of bacteria related'

Line 78: whose? Sentence is confusing as written. For clarity, replace 'whose' with "with an" and insert a "that" after abundance

Changed accordingly

Line 80: PhnWAY not introduced, add "that" after fact

We have now fully introduced the degradation systems in the previous paragraph:

'Unlike the majority of C-O-P monoester bonds, the C-P bond requires specific enzymes to break it, such as the C-P lyase^{26,27}. Several 2AEP-specific phosphonate degradation systems have been characterised (Fig 1A), including the 2AEP transaminase (PhnW) – phosphonoacetaldehyde hydrolase (phosphonatase – PhnX) system^{28,29} and the PhnW – phosphonoacetaldehyde dehydrogenase (PhnY) – phosphonoacetate hydrolase (PhnA) system³⁰⁻³². The phosphonate dioxygenase (PhnY*) phosphohydrolase (PhnZ) system has also been shown to degrade 2AEP^{33,34}, though at least some homologs of this system are specific to (hydroxy-)methylphosphonate and cannot degrade 2AEP^{35,36}. In addition, a gene encoding a recently characterised (*R*)-1-hydroxy-2-aminoethylphosphonate ammonia lyase (PbfA) is often found in *phnWX* and *phnWAY* operons³⁷, expanding the known repertoire of aminophosphonate degrading capabilities³⁷.'

Line 82-83: This statement about transporters being superb molecular tools for investigating in-situ cycling of metabolites, while being a true statement, seems out of place/unconnected.

Agreed. We have now moved this up to the start of the paragraph introducing phosphonate transport systems.

'When analytical methods are not sensitive enough to accurately quantify the concentration and turnover of specific environmental metabolites, screening for the expression of their respective uptake systems becomes an important tool in understanding their *in situ* cycling^{37,40-42}.'

The amended sentence now reads:

'Here, we sought to identify transporters required for 2AEP catabolism in environmental bacteria lacking PhnCDE or PhnSTU.'

Lines 94-97: This sentence is very confusing as written with too many clauses so that it's hard to sort out which clause is related to which statement.

Thank you for this suggestion. We have now amended this as follows:

'In *Pseudomonas putida* BIRD-1 (hereafter BIRD-1), a periplasmic substrate binding protein associated with one of these putative transporters was induced under Pi-deplete growth conditions in a PhoBR-dependent manner⁴³. We hereafter refer to this substrate-binding protein (PPUBIRD1_4925) as 2-aminoethylphosphonate X (AepX). AepX belongs to the same family (pfam13343) as PhnS, iron and sulphate substrate-binding proteins but is....'

Line 101: Other statements reference Pi being exported whenever AEP is the N source (i.e., AEP as just N source or as N and P source). What that not the case in BIRD-1 (or did the authors not check for Pi release when BIRD-1 was growing on AEP as an N source with added Pi? This should be made clearer (as should the later statements).

As per reviewer 2's comment, we have now performed this experiment in *P. putida*. We have amended the line in the results section as well as updating Figure S1.

'BIRD-1 was capable of growth on 1.5 mM 2AEP as either a sole P, sole N or sole N and sole P source, the latter two resulting in mineralisation of excess Pi which was subsequently exported from the cell (Fig 1C, Fig S1).'

As well as:

'However, *S. stellulata* lacks AepP but is still capable of Pi-insensitive growth and Pi export when grown on 2AEP as a sole N or sole N and sole P source (Fig S5A).'

Line 179: N has been introduced as abbreviation for nitrogen

Thank you for spotting this. Amended accordingly

Line 180: What or where is table S3? Is it the "dataset" that is attached? If so, please make that clear and add a table legend so that the reader knows what they're looking at.

This is a table containing the raw data for the proteomics analysis of *Stappia stellulata*. We uploaded it as a separate excel file.

Line 188: Statement as written implies that this is absent from the Rhodobacteriaceae. Do you mean in addition to?

Agreed. Changed to:

'*Alphaproteobacteria* in addition to *Rhodobacteraceae*, marine *Deltaproteobacteria*,.....'

Line 316: What do you mean by co-culture experiments? Please provide a bit more detail.

As spotted also by reviewer 2 this mention of co-cultures was a relic of an earlier version of the manuscript and has now been removed.

Line 325: Table S4 is missing strain information. Also, as a general comment, I assume that all of the strains used in this work were axenic. The authors should explicitly state this somewhere.

Thank you for this suggestion. Gene names and locus tags have been added to Table S4, along with the purpose of each primer. We have added the following sentence to the beginning of the methods section:

'All strains used in this work were axenic.'

Methods: General comment. There are a lot of abbreviations in the methods of things that are somewhat standard/well known to a biologist, but probably should still be spelled out given the audience for Nature Communications (HEPES, LDS, PCR).

Thank you for your suggestion. We have removed all unnecessary abbreviations from the entire manuscript.

Figure 3 Legend: This legend is hard to follow and doesn't explain all of the components of the figure. What do the colors mean? What do the dashed lines mean?

Thank for this comment. We have now added the following sentences at the end of the figure legend.

'Vertical dashed lines represent an LFQ_{Log2} difference >3 or <3 . The horizontal dashed line illustrates a cut off for a significant Q value ($p < 0.05$). Sky blue represents protein showing no significant difference between treatments. Peach and red indicate proteins significantly changing in abundance <3 -fold or >3 -fold, respectively.'

REVIEWERS' COMMENTS

Reviewer #1 (Remarks to the Author):

I want to congratulate and thank the authors for considering the knowledge and experience of the reviewers on this subject, who I believe will be satisfied with the effort made. The manuscript and the study were strengthened by the comments and suggestions made and the additional experiments that were carried out.

Although some concepts remain to be confirmed in the field, I believe that this document lays a solid foundation for further understanding the role of phosphonates in the oceanic phosphorus cycle.

Therefore, I would recommend the publication of this manuscript.

Reviewer #2 (Remarks to the Author):

I'd like to thank the authors for their detailed responses and clarifications, and especially for the additional labwork. I'm fully satisfied that the queries I raised have been addressed, and hope to see the paper published in its final form soon.

Reviewer #3 (Remarks to the Author):

The authors have appropriately addressed my comments in my previous review. The additional data presented in response to my and other reviewer comments have improved an already impressive manuscript. I did note some weirdness with bolding of certain words that were not consistent and had not been in the previous version of the manuscript. I'm not sure if those were just there because of the way they were tracking changes. Also the newly added sentence at the bottom of page 2 (lines 39-41) has abbreviation issues, but all other abbreviation issues appear to have been fixed. I especially appreciate the inclusion of the full names of various 2-AEP phosphonate degradation systems that was added to the introduction.

REVIEWER COMMENTS

Once again, we thank the reviewers for their time helping us improve our manuscript.

Reviewer #1 (Remarks to the Author):

I want to congratulate and thank the authors for considering the knowledge and experience of the reviewers on this subject, who I believe will be satisfied with the effort made. The manuscript and the study were strengthened by the comments and suggestions made and the additional experiments that were carried out.

Although some concepts remain to be confirmed in the field, I believe that this document lays a solid foundation for further understanding the role of phosphonates in the oceanic phosphorus cycle.

Therefore, I would recommend the publication of this manuscript.

Reviewer #2 (Remarks to the Author):

I'd like to thank the authors for their detailed responses and clarifications, and especially for the additional labwork. I'm fully satisfied that the queries I raised have been addressed, and hope to see the paper published in its final form soon.

Reviewer #3 (Remarks to the Author):

The authors have appropriately addressed my comments in my previous review. The additional data presented in response to my and other reviewer comments have improved an already impressive manuscript. I did note some weirdness with bolding of certain words that were not consistent and had not been in the previous version of the manuscript. I'm not sure if those were just there because of the way they were tracking changes. Also the newly added sentence at the bottom of page 2 (lines 39-41) has abbreviation issues, but all other abbreviation issues appear to have been fixed. I especially appreciate the inclusion of the full names of various 2-AEP phosphonate degradation systems that was added to the introduction.

We have amended line 39-41 and two other instances where phosphorus was spell in full, instead of P. These are marked in the final word document with a blue P.

Perhaps the bolding of certain words was a formatting error when converting to .pdf? We have thoroughly checked the word document and cannot find any reference to this.